# A Semisupervised Deep Learning Neural Network Using Pseudolabels for Three-Dimensional Shallow Strata Modelling and Uncertainty Analysis in Urban Areas from Borehole Data

Jiateng Guo1*, Xuechuang Xu1, Luyuan Wang1, Xulei Wang1, Lixin Wu2, Mark Jessell3, Vitaliy Ogarko3, Zhibin Liu1 and Yufei Zheng1

[1]College of Resources and Civil Engineering, Northeastern University, Shenyang, China; 2000985@stu.neu.edu.cn (X. X.); wangluyuan9@163.com (L. W.); wxldbdx@163.com (X. W.); 207158587@qq.com (Z. L.); 15333134708@163.com (Y. Z.)

[2]School of Geosciences and Info-Physics, Central South University, Lushan Nanlu 932, Yuelu District, Changsha 410012, China; awulixin@263.net (L. W.)

[3]Centre for Exploration Targeting / Mineral Exploration Cooperative Research Centre / DARE Centre, School of Earth Sciences, The University of Western Australia, Perth, Australia; mark.jessell@uwa.edu.au (M.J.); vitaliy.ogarko@uwa.edu.au (V. O.)

*Correspondence to: Jiateng Guo (guojiateng@mail.neu.edu.cn; Tel.: +86-24-8368-7693)

**Abstract**. Borehole data are essential for conducting precise urban geological surveys and large-scale geological investigations. Traditionally, explicit and implicit modelling have been the primary methods for visualizing borehole data and constructing 3D geological models. However, explicit modelling requires substantial manual labour, while implicit modelling faces problems related to uncertainty analysis. Recently, machine learning approaches have emerged as effective solutions for address these issues in 3D geological modelling. Nevertheless, the use of machine learning methods for constructing 3D geological models is often limited by insufficient training data. In this paper, we propose the semisupervised deep learning using pseudolabels (SDLP) algorithm to overcome the issue of insufficient training data. Specifically, we construct the pseudolabels in the training dataset using the triangular irregular network (TIN) method. Three 3D geological model is constructed using borehole data obtained from a real building engineering project in Shenyang, Liaoning Province, NE China. Then, we compare the results of the 3D geological model constructed based on the SDLP with those constructed by a support vector machine (SVM) method and an implicit HRBF modelling method. Compared to the 3D geological models constructed using the HRBF algorithm and SVM algorithm, the 3D geological model constructed based on the SDLP algorithm better conforms to the sedimentation patterns of the region. The findings demonstrate that our proposed method effectively resolves the issues of insufficient training data when using machine learning methods and the inability to perform uncertainty analysis when using the implicit method. In conclusion, the semisupervised deep learning method with pseudolabelling proposed in this paper provides a solution for 3D geological modelling in engineering project areas with borehole data.

## 1. Introduction

Three-dimensional (3D) urban geological models are digital representations of subsurface strata and their associated features (Houlding, 1994). In recent years, the utilization of 3D geological models has expanded across various geological fields, such as mineral exploration (Zhang et al., 2021), geological storage (Thanh et al., 2019), groundwater resource estimation (Thibaut et al., 2021), geological disaster early warning generation (Høyer et al., 2019; Livani et al., 2022), and engineering geological condition evaluation (Chen et al., 2018; Guo et al., 2021; Lyu et al., 2021; Marz´an et al., 2021).

The commonly used 3D geological modelling data include borehole data, geophysical data, survey and mapping data, and outcrop data. Among these, borehole data provide the most accurate reflection of subsurface geological information (Guo et al.,2022). Notably, 3D geological modelling from borehole data can be divided into explicit modelling and implicit

modelling (Jessell, 2001; Caumon et al., 2007; Wang et al., 2018). The explicit modelling approach can be used to directly delineate geological formations and interpret tectonics based on borehole data. Explicit 3D geological modelling methods are widely used in the 3D modelling of mines and regional geological structures, and they include the interactive 3D forward modelling method (Yang et al., 2011), generalized tri-prism (GTP) modelling method (Wu et al., 2004; Che et al., 2009) and parametric surface method (Lyu et al., 2021). However, these approaches rely heavily on the expertise of geologists and often prove time-consuming and labour-intensive when dealing with large-scale borehole data.

Implicit modelling methods are used to construct a 3D geological model by establishing the implicit equation of the isosurface representing the geometric shape of a geological body and using a series of implicit function visualization methods (Jessell M. et al., 2022). In other words, a complex 3D geological object is represented as a continuous function of geological coordinates (Wang G.W. et al., 2011; Zhong D. Y. et al., 2021). This method does not require extensive human–computer interaction and has the advantages of high modelling accuracy, excellent smoothness and high spatial analysis efficiency (Sun H. et al., 2023). It is widely used in the field of geological modelling (Hillier M. J. et al., 2014; Calcagno P. et al., 2008; Shi T. D. et al., 2021) and provides results to complement the results of most urban geological surveys (de la Varga M. et al., 2019). Common implicit modelling methods include nearest neighbour value interpolation (Olivier R. et al., 2012), inverse distance weighted (IDW) interpolation (Liu H. et al., 2020; Liu Z et al., 2021), discrete smooth interpolation (DSI) (Mallet J. 1997), kriging (Wang G.W. et al., 2011; Thanh H. V. et al., 2019), the moving least squares (MLS) method (Manchuk J. G. et al., 2019), and the radial basis function (RBF) method (Caumon G. et al., 2013; Hillier M. et al., 2014; Cuomo S. et al., 2017; Martin R. et al., 2017; Skala et al., 2017; Zhong D. Y. et al., 2019).

The sparsity of borehole data, the complexity of geological bodies or geological phenomena, and the limitations of human cognition and expression lead to uncertainty in the relationship between the geometric form of a 3D geological model and the corresponding geological system (Caumon et al., 2007; Caers, 2011; Pakyuz-Charrier et al., 2018; Guo et al., 2022). When using the implicit modelling method to construct a 3D geological model, an implicit function can only correspond to one kind of geological interface expression. The construction of 3D geological models by establishing implicit equations cannot effectively address this uncertain relationship. Fortunately, the machine learning method is a kind of stochastic modelling method which can generate many possible geological models from one borehole dataset, and easily perform uncertainty analysis by using information entropy or confusion index, etc. Therefore, this paper introduces a new geological modelling method based on machine learning approaches to evaluate the accuracy of the generated model by uncertainty analysis.

Machine learning methods have been widely used in 3D geological modelling, and they are generally applied in unsupervised or supervised 3D geological modelling (Wang et al., 2023). Unsupervised machine learning algorithms (e.g., k-means clustering, self-organizing maps, and Gaussian mixture models) can be used to translate multisource geophysical datasets into 3D lithological models by measuring the similarity between properties in feature space (Hellman et al., 2017; Giraud et al., 2020; Whiteley et al., 2021; Zhang et al., 2022). Supervised machine learning algorithms (e.g., random forests and artificial neural networks) can be applied to construct 3D lithological models by training from labelled geophysical and geological datasets (Jia et al., 2021; Lysdahl et al., 2022). Despite obtaining encouraging results with supervised machine learning algorithms, most studies have not addressed the following critical challenges regarding supervised machine learning algorithms for 3D geological modelling:

(1) In the field of 3D geological modelling, precise and adequate geological investigating data will help generate more accurate subsurface representations. However, due to the high exploration cost, borehole data which can precisely reveal relationships between stratigraphy and tectonic features in a study area are usually limited. Utilizing the precise information obtained via boreholes as labelled data may not be enough to predict many unknown areas. The correctness of the results predicted by machine learning still requires further research.

(2) The labelled geological datasets are mainly composed of borehole data from early exploration phases (Jia et al., 2021; Lysdahl et al., 2022). The number of lithological sample categories in drilling datasets is commonly imbalanced. A classification dataset with skewed class proportions can influence the performance of machine learning algorithms (Chawla et al., 2002; Batista et al., 2004). However, very little published research has addressed the sample imbalance issue in the context of training supervised machine learning algorithms for 3D lithological modelling.

Compared with machine learning methods, deep learning algorithms improve the ability to learn from mining data and are often combined with complex geophysical and geochemical data for modelling. Currently, there is a wealth of research on neural network-based deep learning methods for addressing geological issues such as tectonic recognition (Titos et al., 2018), mineral identification and classification (Xu and Zhou, 2018), and seismic data inversion (Huang et al., 2020) . Furthermore, in the realm of constructing 3D geological models, deep learning approaches using neural networks have also gradually garnered significant attention from numerous scholars (Laloy et al., 2017; Zhang et al., 2019; Ran et al., 2020; Zhang et al., 2018; Michael Hillier et al., 2021, 2022; S Avalos and Ortiz, 2020). However, the issue of insufficient training data has yet to be adequately addressed.

In this paper, we propose a semisupervised deep learning using pseudolabels (SDLP) algorithm for constructing 3D geological models. The algorithm is used to overcome the problems of a lack of accurate labelled data in machine learning methods and the inability of implicit modelling methods to perform uncertainty analysis. The shallow borehole data obtained from the real engineering project Shenyang, Liaoning Province, are used to construct 3D geological models via the proposed algorithm. To demonstrate the applicability of the SDLP algorithm, the accuracy, precision, recall, and F1 score results of the SDLP algorithm are compared with those of a classic support vector machine (SVM) algorithm based on a test dataset. To further assess the accuracy of the SDLP, the profiles of the 3D geological models constructed by the SDLP, SVM, and herite radial basis function (HRBF) are compared. The findings indicate that the SDLP algorithm can effectively solve problems where uncertainty analysis cannot be performed via the implicit modelling method, and can solve the problem that lack of training datasets by pseudolabels.

## 2. 3D Modelling Method Based on Deep Learning

### 2.1. Borehole data preprocessing

Table 1. The average thickness, maximum thickness, minimum thickness and frequency of occurrence of the different strata

|  | Frequency | Average (m) | Maximum (m) | Minimum (m) |
|---|---|---|---|---|
| fill | 167 | 1.14 | 4.1 | 0.4 |
| Clay-1 | 128 | 2.21 | 6 | 0.7 |
| Clay-2 | 58 | 3.46 | 9.8 | 0.5 |
| Clay-3 | 107 | 5.94 | 12.8 | 0.5 |
| Clay-4 | 54 | 2.86 | 5.8 | 0.5 |
| Sand-1 | 25 | 3.34 | 8.1 | 1.2 |
| Stone-1 | 71 | 6.30 | 14 | 1.3 |
| Stone-2 | 104 | 3.91 | 10 | 0.5 |
| Stone-3 | 72 | 6.22 | 12.5 | 1.2 |
| Residual-1 | 52 | 10.98 | 16.1 | 4.8 |
| Residual-2 | 50 | 4.77 | 13.8 | 2 |
| Residual-3 | 44 | 5.47 | 13.9 | 1 |

A total of 167 boreholes obtained from a real engineering project in Shenyang city were used to build the 3D geological model in this study. The primary objective of the project is to ensure building stability. These boreholes are distributed in a 305×264 m area, with an average spacing of approximately 23 metres between adjacent boreholes. The average depth of the boreholes is 29.5 metres. The minimum thickness of the formations revealed by the boreholes is 0.4 metres, and the

maximum thickness is 16.1 metres (Table 1). The original borehole data mainly include borehole coordinates (X, Y), elevation, lithological thickness, lithological bottom depth, borehole number, lithological ID, etc.

This paper uses deep learning methods for 3D geological modelling, which can further simplify the modelling problem into a strata classification problem. In this method, the coordinate data and strata depth data obtained from boreholes are used as input vectors, and the lithological attributes of the boreholes are used as output vectors. In this study, the borehole data were simplified into continuous one-dimensional data when creating the dataset. However, there are significant differences in the lengths and frequencies of different formations within the borehole dataset (Table 1). For example, in terms of formation

thickness, the maximum thickness is 16.1 m, while the minimum thickness is only 0.4 m. In terms of the formation occurrence frequency, the most frequent label, "fill," occurs 167 times, while the least frequent label, "sand-1," occurs 25 times. This significant difference may lead to overfitting of the training model and ultimately result in poor training performance. Therefore, preprocessing of the borehole data is needed. An upsampling method is proposed to avoid overfitting in the training model caused by imbalanced training datasets in this study.

Based on the above discussion, an unequal interval sampling method is adopted in this paper (Fig. 1). In the figure, $H_{11}$-$H_{35}$ represents unequal-interval sampling for each stratum in the borehole, while $H_{11}P_1$-$H_{35}P_5$ represents unequal-interval sampling for each stratum in the deterministic section. Compared with equal-interval sampling, unequal-interval sampling involves changes in the sampling interval according to the thickness of different strata, thereby ensuring the balance of the sampled data. Otherwise, thinner strata may be difficult to predict or deemed to be outliers due to insufficient sampling. As

shown in Fig. 1, different colours in the borehole region represent different strata attributes, and the strata data are displayed in strips that are continuously distributed in the vertical direction. The attributes of a single stratum are continuously unique within the corresponding depth interval, and there are no data gaps between strata.

    Due to the high reliability of borehole data, these data can be directly or indirectly used for the generation of accurate models. By applying the Delaunay principle to borehole position points, a surface triangular irregular network (TIN) is

135 created. The TIN is a method used for two-dimensional spatial data modelling and analysis in geography. This TIN encompasses the fundamental topological relationships between adjacent boreholes. If the stratum attributes of two neighbouring boreholes within each TIN are similar, they are connected to form a deterministic section. To ensure accurate geological predictions and eliminate the influence of distant and loosely correlated borehole connections, narrow triangles are removed from the TIN. The threshold for determining whether a triangle is a narrow triangle based on the measurement

of its smallest angle is set to 20 degrees. This approach, similar to the generalized tri-prism (GTP) model, preserves the internal connectivity among the three corresponding boreholes and enables the simulation of various complex geological phenomena. Once the deterministic sections are connected, unequal interval sampling is conducted both horizontally and vertically, and the sampling density at the borehole locations is balanced to avoid overly dense sampling that may impact network training. The unequal interval sampling formula for borehole data is expressed as Equation (1), and the unequal

interval sampling point coordinate formula for deterministic sections is expressed as Equation (2).

$$Z_{ij} = \frac{(S_{ij} - S_{ij-1})}{n} \tag{1}$$

$$\begin{cases} P_{ijx} = x_1 + \frac{x_2 - x_1}{n}(2j-1) \\ P_{ijy} = y_1 + \frac{y_2 - y_1}{n}(2j-1) \\ P_{ijz} = \frac{D_1 C_2 + A_1 C_2 P_{ijx} + B_1 C_2 P_{ijy} - D_2 C_1 - A_2 C_1 P_{ijy} - B_2 C_1 P_{ijy}}{C_1 C_2 n}(2i-1) \end{cases} \tag{2}$$

where $S_{ij}$ is the bottom depth of the jth stratum in the ith borehole, n is the number of samples from each stratum, and $Z_{ij}$ is the sampling interval of the jth stratum in the ith borehole. $P_{ijx}$, $P_{ijy}$, and $P_{ijz}$ represent the x, y, and z coordinates of the

150 sampling point in the ith row and jth column of a section, respectively. $x_1$, $y_1$, $x_2$, and $y_2$ are the coordinates of the two

connected boreholes in a section. $A_1$, $B_1$, $C_1$, $D_1$, $A_2$, $B_2$, $C_2$, and $D_2$ are the parameters of the straight-line equations representing the top and bottom boundaries of the strata for the connected boreholes.

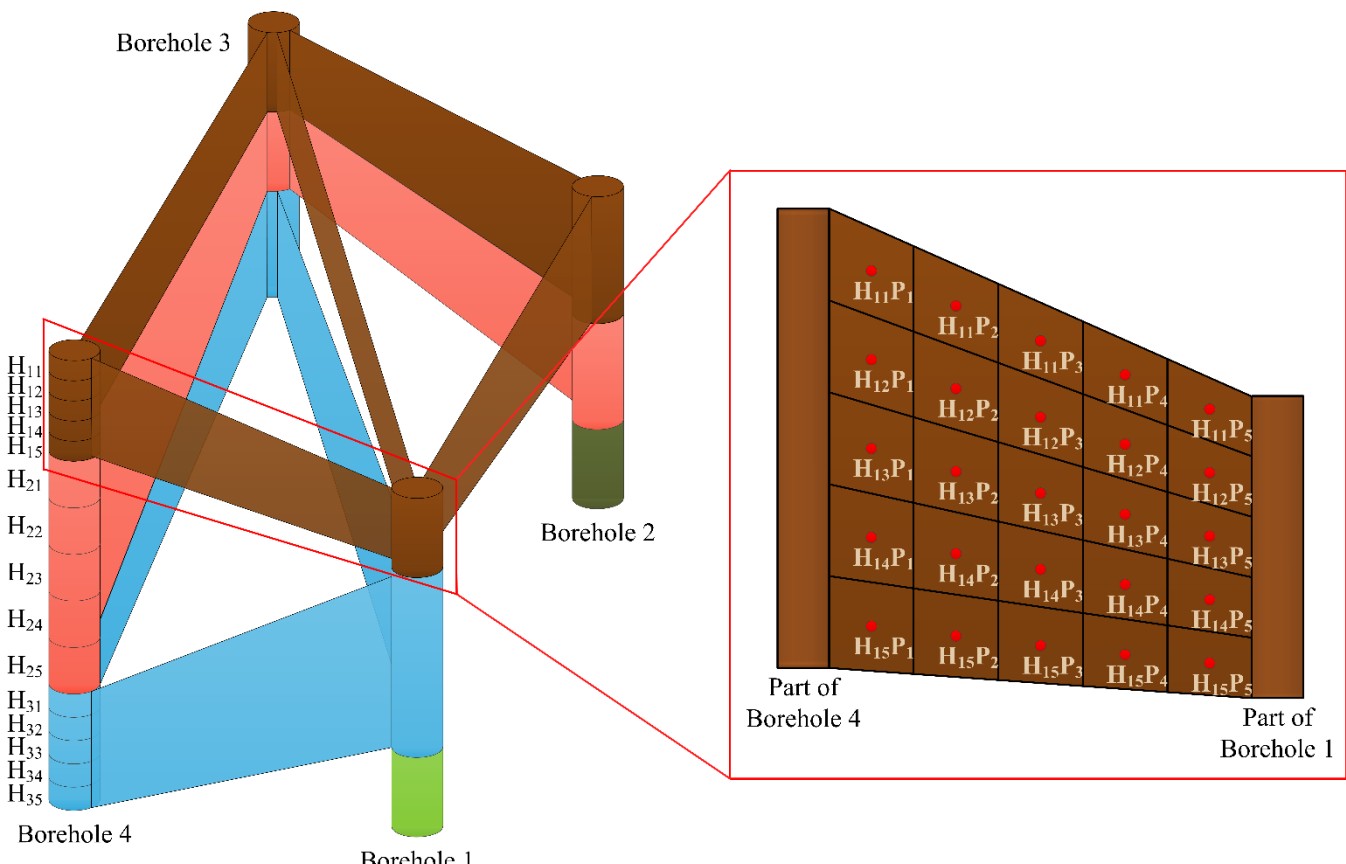

**Figure 1. Resampling of borehole data. Upsampling on the boreholes (left); upsampling on the deterministic sections (right).**

The difference in the number of digits between coordinate data (typically 7-8 digits with 3 decimal places) and stratum depth (typically 1-2 digits with 1 decimal place) in borehole data can lead to numerical computation issues in computer systems, making it difficult to train the model and adjust parameters, ultimately affecting the training results of the model. After performing data normalization based on the raw data, each indicator is scaled to a specific range, allowing for comprehensive comparative evaluation. To eliminate the influence of digit disparity among input features, ensure the equal impact of different features on model training, and achieve convergence, it is necessary to apply min–max normalization to the data and map the resulting values to the range of 0 to 1. For any dataset x, the mapping function is as follows:

$$x' = \frac{x - x_{min}}{x_{max} - x_{min}} \tag{3}$$

where $x_{max}$ is the maximum value of the sample data and $x_{min}$ is the minimum value of the sample data. x' is the normalized result, and x is the input of the model data. Through this normalization method, the convergence speed of the network training model is improved, the training accuracy is improved, and model training becomes easier.

## 2.2. Construction of deep neural networks

A multilayer perceptron (MLP) is a feedforward artificial neural network that learns to form certain rules through training based on input and output indicators. Thus, the results closest to the expected output are obtained after inputting certain values. An MLP is a multilayer feedforward neural network based on the backpropagation algorithm. Each unit between layers in an MLP has a weight with an initial preset value, and unit training is performed using the backpropagation algorithm to adjust the weights between hidden layers. The input data are output after passing through multiple hidden layers and compared with the expected labels to obtain the corresponding error, which is then propagated layer by layer backwards to adjust the weight of each layer. After multiple adjustments, suitable weights for the model are obtained. The relationship between layers can be expressed as shown in Equation (4): In the network model, the coordinates of each upsampled spatial point in the prediction area, x, y, and z, are used as inputs, and the geological properties of the spatial points are output. Each input represents a spatial feature dimension, and through four fully connected layers, the input data are processed and transformed. Each hidden layer contains multiple nodes, where each node is connected to all nodes in the previous layer. By multiplying by weights and applying an activation function, the input undergoes nonlinear transformation, resulting in expanded dimensionality. This result encompasses the deep features of the sample, and samples of different categories should have different high-dimensional features. The number of neurons in the hidden layer varies according to the complexity of the model, and the rectified linear unit (ReLU) activation function is used between hidden layers. To prevent overfitting, a dropout function is added to the penultimate fully connected layers of the network to randomly reduce the number of neurons. The dropout percentage is set to 10%. Finally, the output value of each category is normalized using the exponential function through a fully connected layer and a softmax layer, and the sum of the probabilities of all categories is 1. The predicted results of each data point are integrated to form the entire 3D geological model (Fig. 2). The network model uses the Adam optimizer, and the loss function adopted is the cross-entropy loss function, which is commonly used in multiclassification tasks. The detailed parameters of the deep neural networks are shown in Table 2.

Table 2. The network architecture and parameters of the deep neural networks in this paper

| Parameters | Value |
|---|---|
| Training set: Validation set: Test set | 6:2:2 |
| Number of hidden layers | 4 |
| Hidden layer 1 | 128 |
| Hidden layer 2 | 256 |
| Hidden layer 3 | 512 |
| Hidden layer 4 | 1024 |
| Learning rate | 0.003 |
| Activation function | ReLU |
| Number of training epochs | 2000 |
| Loss function | Cross entropy loss |
| Optimizer | Adam |

$$Y_j = \sum_{i=1}^{n} W_{ij} X_i + b \tag{4}$$

where $Y_j$ is the input of the next layer, $W_{ij}$ is the connection weight from cell $X_i$ of the previous layer to cell $Y_j$ of the next layer, and b denotes the offset value.

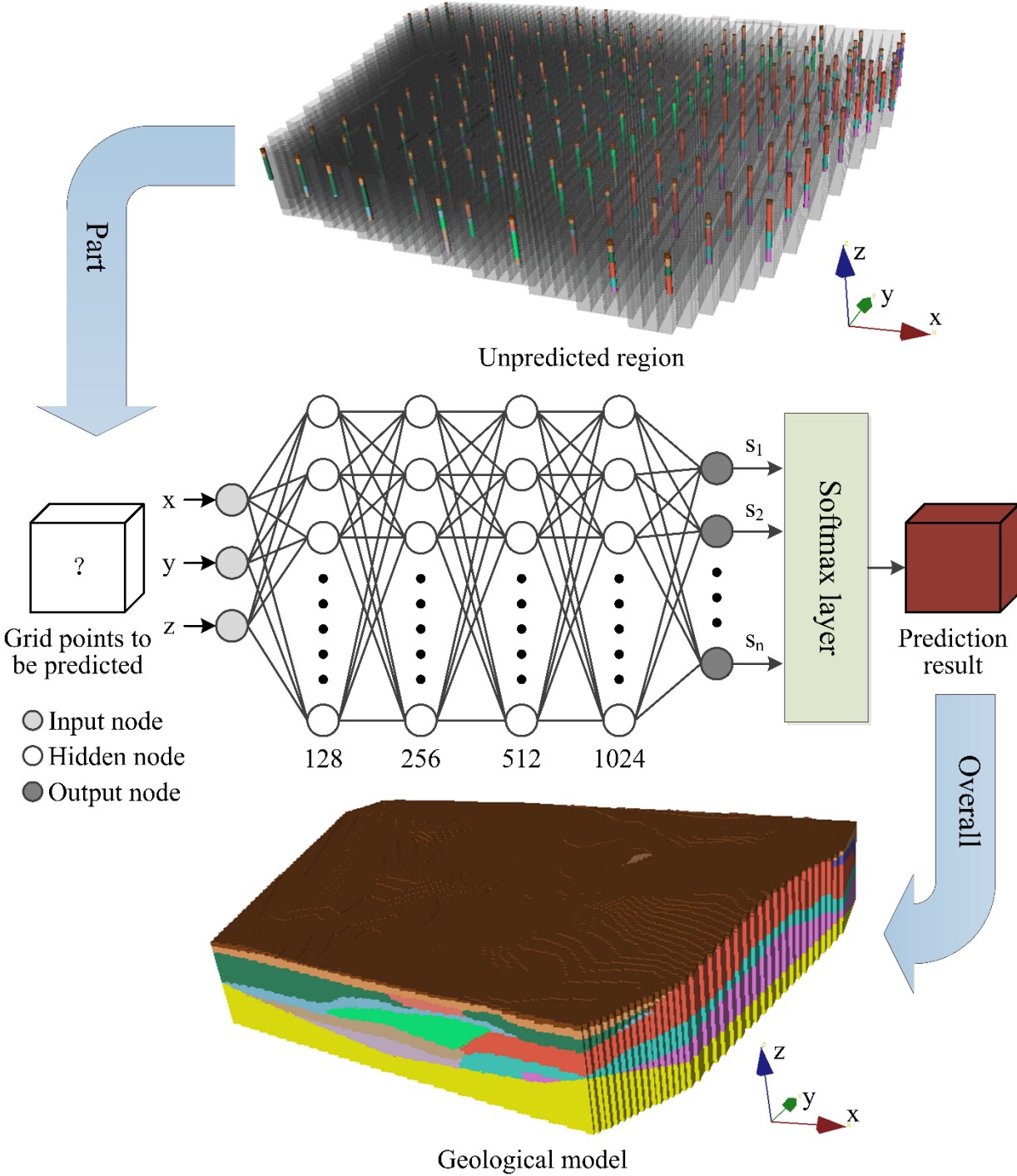

 **Figure 2. Architecture of a deep neural network. Light grey nodes are input features, dark grey nodes are target outputs, and white nodes are internal network nodes.**

## 2.3 Semisupervised deep learning algorithm using pseudolabels

Compared with data from images, point cloud data, etc., borehole data exhibit clustering characteristics with local concentrations but overall dispersion. Due to the large amount of missing point data between boreholes, it is difficult to accurately express the changing features of stratigraphic boundaries and inclination angles. Supervised learning depends on a

large quantity of labelled data to enhance model performance. The labelled data used for training 3D geological models are obtained by upsampling limited borehole points and deterministic borehole profiles. Labelled data associated with spatial grid points in urban areas, which require high modelling precision, are scarce and contain very few features. To effectively solve the labelling problem, semisupervised learning is combined with deep learning, and a model is constructed using a small amount of labelled data and a large amount of unlabelled data with pseudolabels for prediction. This approach is beneficial for expanding the training data.

The attributes of strata are difficult to determine based on a single mathematical formula. Based on the topological relationships established with the TINs of three boreholes, three prisms are constructed using a method similar to the GTP approach by connecting the boreholes based on their stratigraphic properties, and the stratigraphic properties of the interior grid points of the prisms are obtained. For the predicted grid points within the prisms, it is assumed that their stratigraphic properties are similar to the properties of the prism, and when adding pseudolabels, it is assumed that the confidence level for each predicted stratigraphic property is high. Based on this approach, a semisupervised learning method based on pseudolabels is used to generate pseudolabels for the unlabelled data and improve learning performance. First, the model is trained using labelled data. When the model reaches an accuracy of 90% after being trained for a certain number of rounds, the trained model is utilized to predict unlabelled data, and high confidence predictions are selected as pseudolabels. The pseudolabelled data and labelled data are combined and used in training for a certain number of rounds. The above process is repeated until the proportion of newly added pseudolabelled data in each round is lower than a certain threshold. At this point, high-confidence labels areobtained, and the model has been sufficiently trained on all the data.

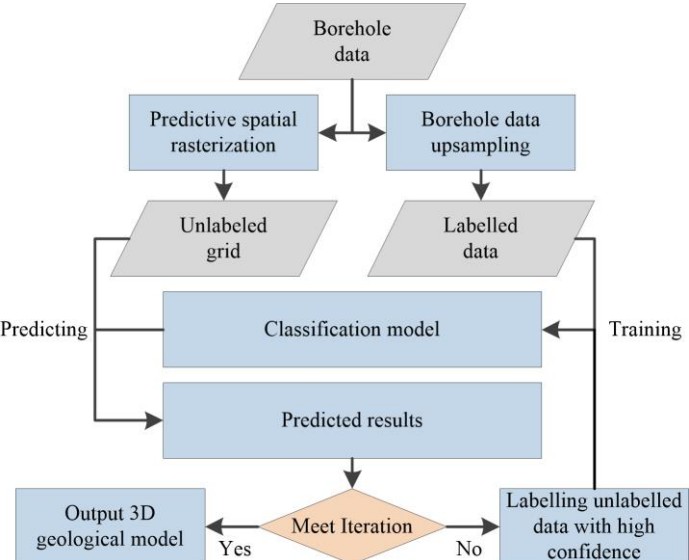

Figure 3. Algorithm flow chart.

## 2.4. Analysis of model uncertainty

The last layer of the neural network classifier normalizes the probability of the output through the softmax layer, and the softmax normalized result can be approximated as the probability corresponding to each stratum at a given data point. Therefore, when analysing the uncertainty of each data point in the raster model, the normalized information entropy can be

introduced to quantitatively evaluate the uncertainty of the geological model. The normalized information entropy formula is as follows:

$$H(X) = -\frac{\sum_{x \in S} p(x) \ln(p(x))}{S_{max}} \qquad (5)$$

where S is the number of possible geological attributes for each data point, $S_{max}$ is equal to ln(n), and n is the number of possible geological attributes. The information entropy of each data point is obtained by calculating the probability p(x) of each data point over all geological attributes. The magnitude of information entropy reflects the degree of complexity at a certain location in the geological model. The closer the information entropy is to 0, the higher the certainty of a data point for a certain stratum attribute, and the closer the information entropy is to 1, the higher the uncertainty of a data point for multiple geological attributes.

In addition, the data can be analysed based on an estimated confusion index (Burrough et al., 1997), and the ambiguity of classification can be evaluated by selecting the results of the two prediction categories with the highest probability for each data point. The confusion index formula is as follows:

$$CI = [1 - (\mu_{max} - \mu_{max-1})] \qquad (6)$$

where $\mu_{max}$ is the probability of the class with the highest predicted probability and $\mu_{max-1}$ is the probability of the class with the second highest predicted probability. CI values range from 0-1 to indicate the degree of confusion predicted for a certain data point, with 0 indicating that a classification result with a low confusion index is not ambiguous and 1 indicating that a classification result with a high confusion index is highly ambiguous.

## 3. Experimental method and verification

The Shenyang city 3D geological models were built using the SDLP, SVM, and HRBF algorithms. All test experiments in this chapter were performed on the same device: an Intel(R) Core (TM) i7-10750H CPU @2.60 GHz with an NVIDIA GeForce RTX 2060, 16.0 GB RAM, and Windows 10 (64-bit).

The ReLU function was used as the activation function in the SDLP algorithm, the initial learning rate was set to 0.001, and the training batch size was set to 512. When the model training accuracy reached 90% or after 500 epochs, the unlabelled grids were labelled with pseudolabels. When the newly added pseudolabels accounted for less than 10% of the number of grids lacking labels in a given epoch, the model was trained for a total of 2000 epochs more before stopping. The training accuracy and loss values are shown in Fig. 4. The accuracy, precision, recall, and F1 score of the SVM, SDLP, and DL (the neural networks are the same as the SDLP but without pseudolabels) algorithms for the test dataset are shown in Table 3.

In the training process, when the labelled data and pseudolabelled data are fused, the boundaries of the stratigraphic categories are finely delineated, the final model training accuracy is above 95%, the loss function is close to zero, and the precision of the model for the test set is 98.16%. A confusion matrix is obtained from the test set (Fig. 5), which reflects the reliability of the evaluation results of the model. The classification accuracy of the model is high for all layers. Some strata are more likely to be confused because they are thin and display similar boundaries as other strata or because the influence of geological phenomena, such as depositional termination. The receiver operating characteristic (ROC) curve is another performance indicator that reflects the performance of a binary classification model in the positive class and thus can be used to evaluate the diagnostic ability of a classifier according to the threshold change (Fawcett, 2006). The area under the ROC curve (AUC) (Fig. 6) represents a comprehensive measure of all possible classification thresholds. AUC values greater than 90%, ranging 75-90%, ranging 50-75% and less than 50% are considered to represent excellent, good, poor and unacceptable performance, respectively (Ray et al., 2010). The area under the curve (AUC) values of the model are all above 90%, indicating that the classification performance of the model is excellent.

Table 3. The accuracy, precision, recall, and F1 score values for the SVM and SDLP algorithms based on the test dataset

|      | Accuracy | Precision | Recall | F1 score |
|------|----------|-----------|--------|----------|
| SVM  | 0.955    | 0.948     | 0.940  | 0.944    |
| SDLP | 0.982    | 0.983     | 0.980  | 0.982    |
| DL   | 0.973    | 0.967     | 0.968  | 0.968    |

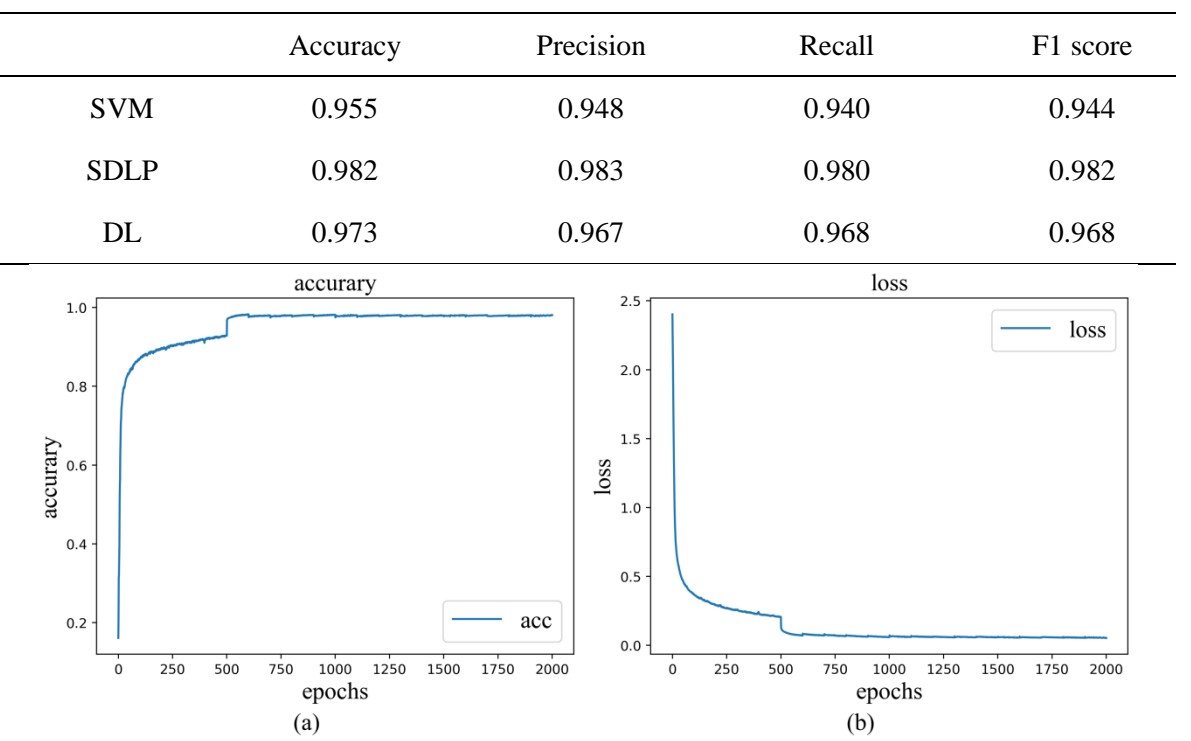

(a)  (b)

**Figure 4. Model training accuracy and loss variation curves**

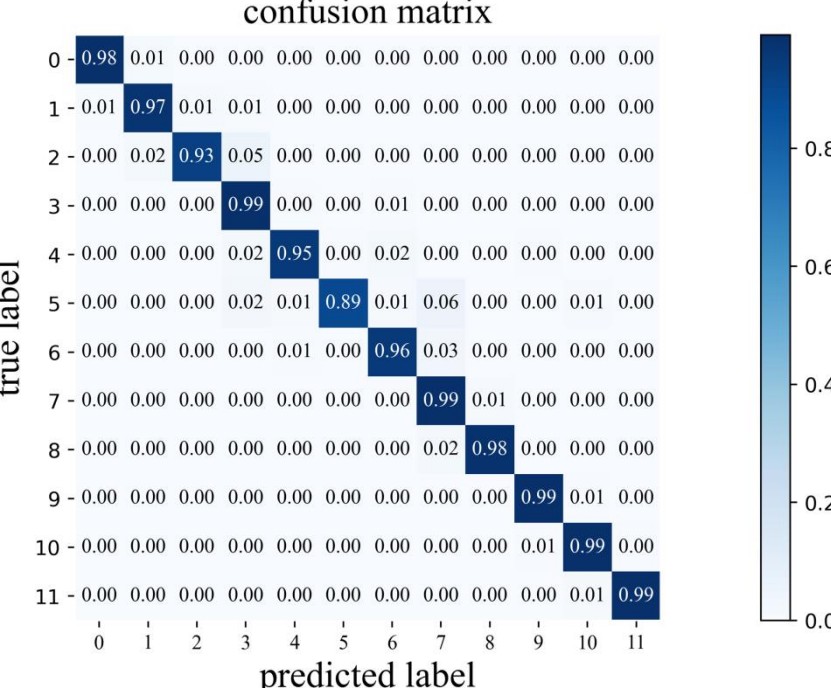

**Figure 5. Confusion matrix of the classification results when the model is applied to the test dataset**

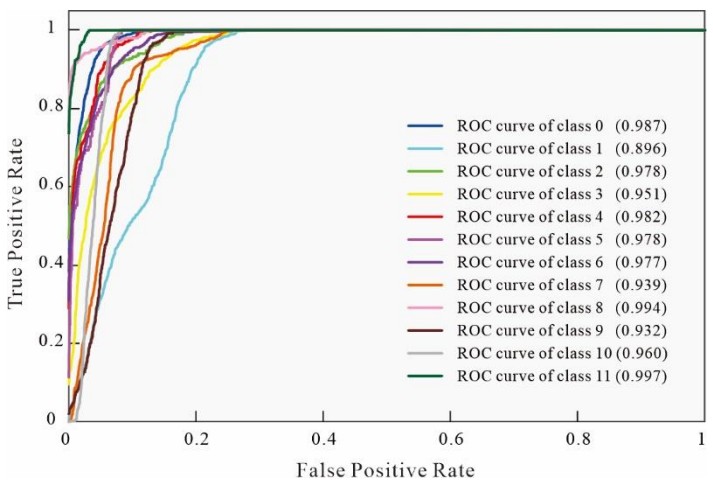

**Figure 6. ROC curve for classification**

The grid used in modelling is 1.5 m × 1.3 m × 0.3 m. The model uses the Tin mesh constructed from the top of boreholes to restrict the surface. The modelling range is determined according to a convex hull built by the borehole data, and the base of the model is determined according to a convex hull built by the bottoms of borehole data. Fig. 7 shows the modelling results for the study area. The model reveals the coverage relationships among the strata and reproduces the contact relationship between the depositional termination and unconformity of the strata.

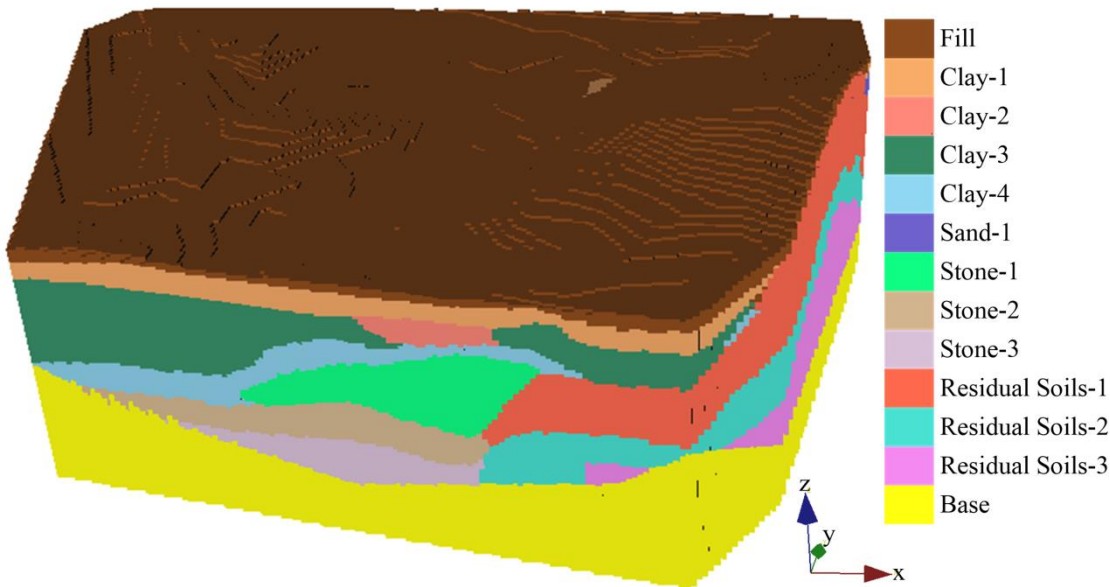

**Figure 7. Model built using deep neural networks and the model legend.**

To test the estimation accuracy at nonborehole locations using the proposed method, the borehole data were divided into a training set and a test set through k-fold cross validation. Learning was performed with the training set of borehole data, and the test set accuracy was compared and analysed, where K was set to 10.

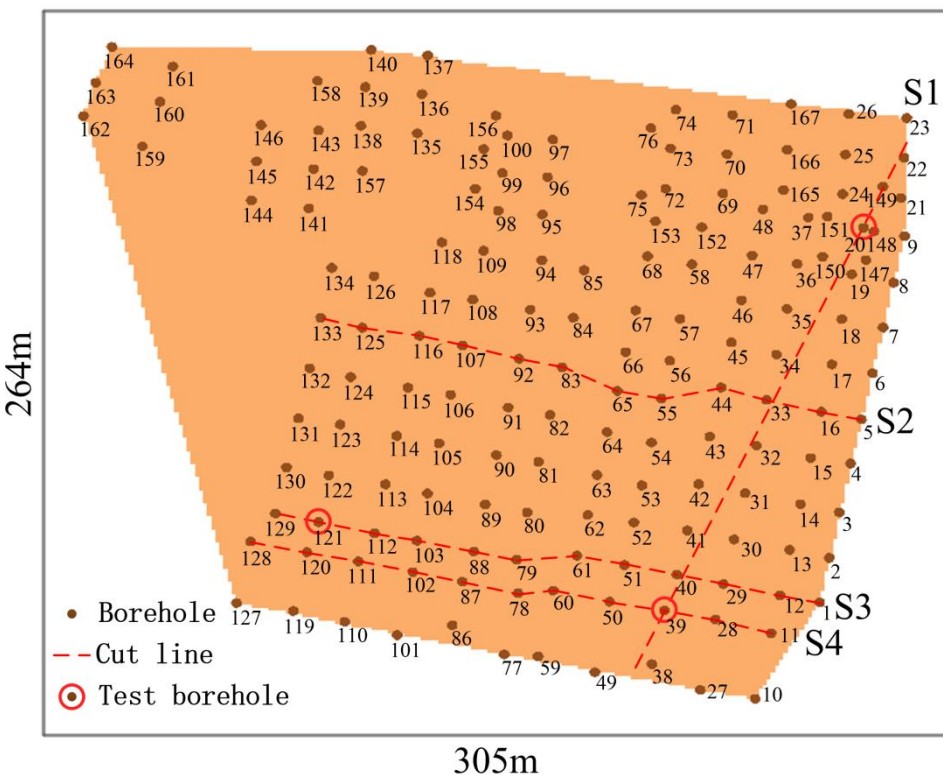

**Figure 8. Borehole distribution and experimental analysis based on different profiles. The red dotted lines are the profiles, and the**
**borehole points circled in red correspond to the boreholes tested using K1**

     The boreholes in the test set were sampled at equal intervals to determine the data point attributes at the boreholes, and the average accuracy of k-fold cross validation was calculated to be 71.65%. Due to the varying amount of geological

information contained in individual borehole data, the importance of different boreholes in constructing the 3D geological model also differs. For instance, test borehole data contain valuable lens body stratigraphic information and stratigraphic extinction information (Fig. 9). Removing the test borehole data would significantly decrease the accuracy of the prediction results. Therefore, we utilize the surface irregular triangulation method generated by the Delaunay rule to determine the topological relationships between boreholes. Based on this approach, boreholes containing a significant amount of geological information are not excluded during K-fold validation. These operations improved the accuracy of K-fold validation from 71.65 to 85.9.

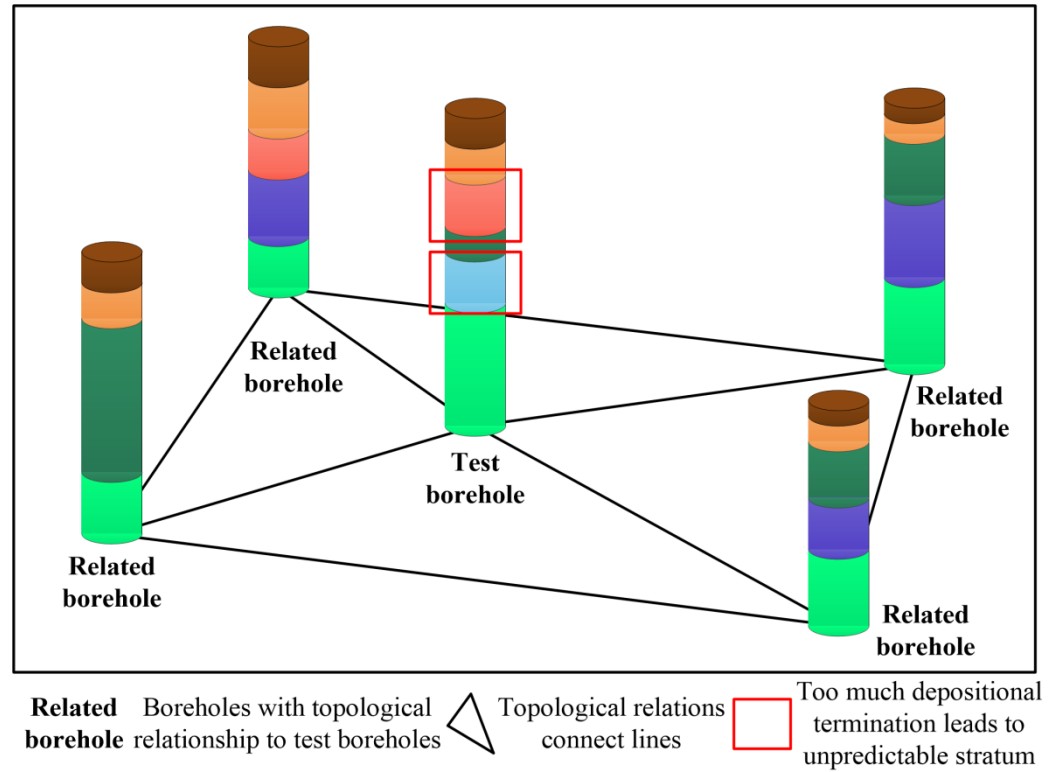

Figure 9. A situation in which too much depositional termination affects the prediction. A related borehole is a borehole that has a topological relationship with the predicted borehole. The red solid frame is the stratum, which is difficult to predict due to the excessive occurrence of depositional termination.

To further analyse the influence of accuracy on the model, a model with complete borehole data and a model with excluded sample K1 test borehole data were established, and the sections of the models through a test borehole were compared (Fig. 10). Fig. 10 shows the results for a straight line through the S1 and S3 profiles. Most of the sections at the boreholes in the test set are consistent with the sections built by a complete borehole. Since some test set boreholes are near depositional terminations, there is a certain difference between the model and the data from test boreholes, but the results are close and reasonable. In summary, the SDLP method displays good prediction ability for neighbouring boreholes and can reveal the distribution characteristics of the strata.

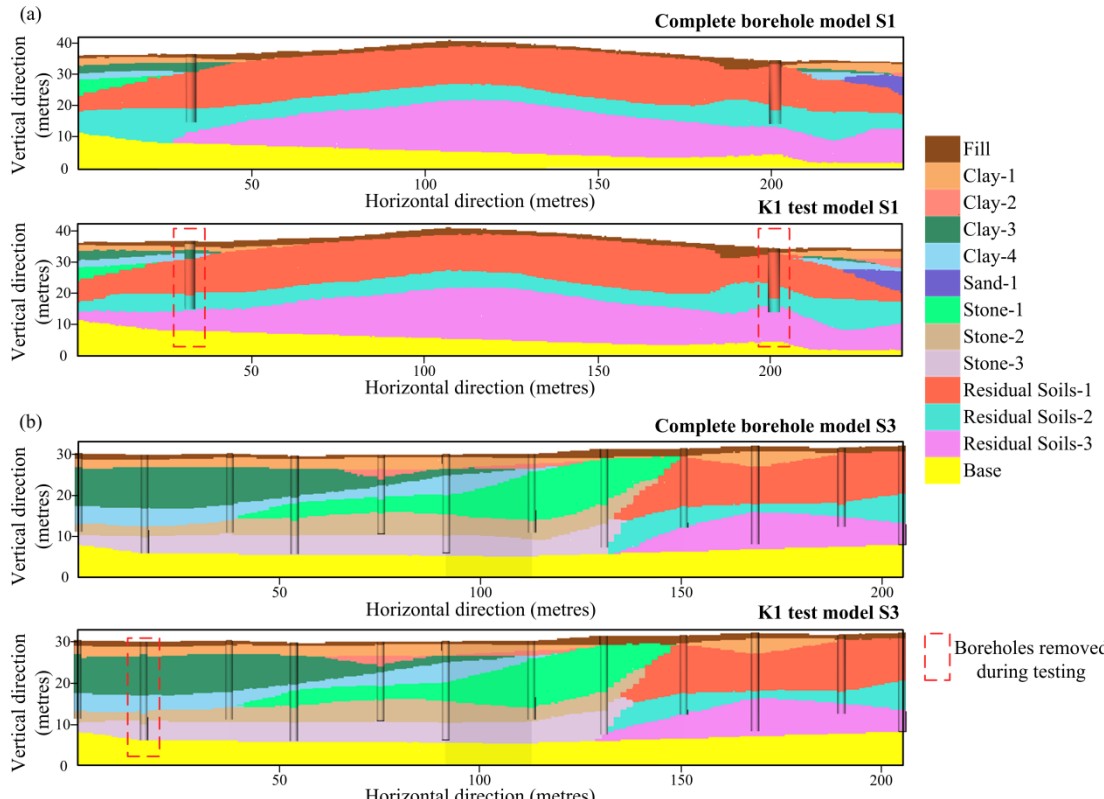

**Figure 10. Comparison of the modelling results for sample K1 with the complete drilling results. The dotted box shows the boreholes considered during the test.**

## 4. Discussion

### 4.1 Verification of the accuracy of the HRBF method

Three-dimensional geological modelling based on the Hermite radial basis function (HRBF) is an implicit function modelling method, and implicit modelling methods based on the HRBF have been widely used in the modelling of ore bodies, regional geological surveys (Guo et al., 2016), urban geological surveys (Guo et al., 2021), tunnelling projects (Xiong et al., 2018), and volcanic formations (Guo et al., 2020). Therefore, in this paper, the HRBF method is used to build a 3D geological model of Shenyang city, and this model is used to compare the accuracy of the SDLP and SVM algorithms. Before evaluating the accuracy of the two algorithms mentioned earlier, it is essential to conduct an accurate analysis of the 3D geological model constructed using the HRBF method. To demonstrate the accuracy of this approach, we first use the HRBF method to build a 3D geological model of Shenyang city. S1, S2, S3, and S4 are profiles within the 3D geological model of Shenyang city and contain many geological strata and complex geological relationships. The accuracy of these profiles can effectively reflect the accuracy of the HRBF modelling method. In the S1 geological profile, the stratigraphic boundaries contained in the borehole dataset nearly perfectly correspond to the boundaries of the three-dimensional geological model built based on the HRBF method (Fig. 11). This matching effect is also demonstrated for the S2, S3, and S4 geological profiles. The accurate correspondence between the borehole data and the cross-sections of the 3D geological

model indicates the precision of the HRBF modelling method in constructing the 3D geological model (Fig. 11b-e).

Furthermore, 3D geological models of Shenyang city built using the HRBF method have been verified to be effective in engineering applications (Guo et al.,2021). In conclusion, the 3D geological model built using the HRBF method can serve as a standard for evaluating the quality of 3D geological models constructed with the SDLP and SVM algorithms.

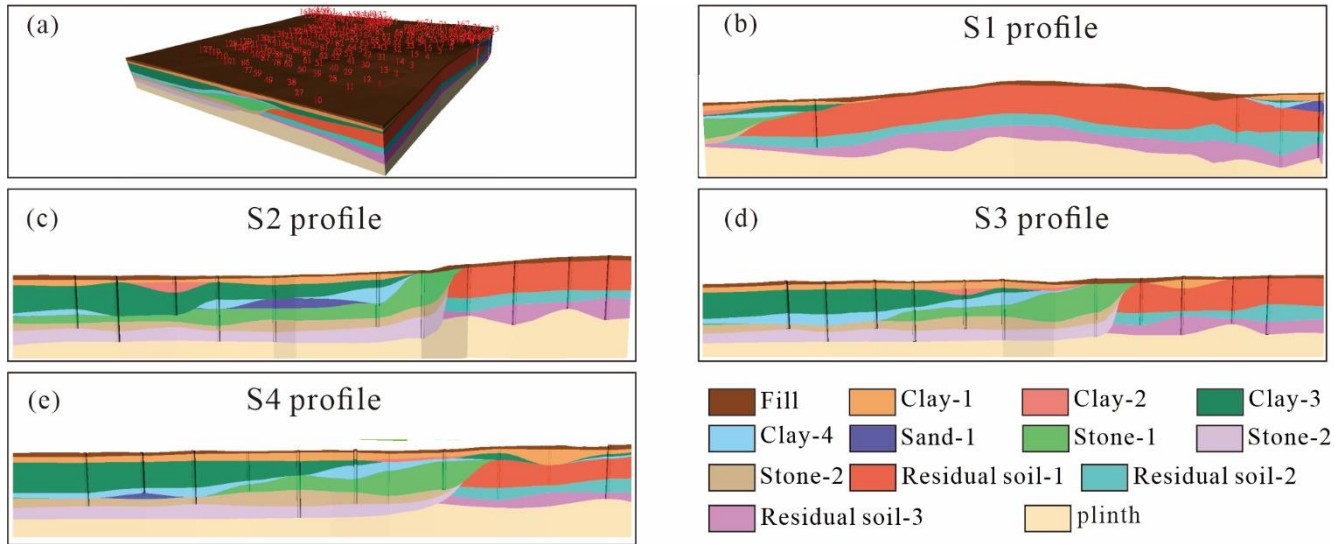

**Figure 11 (a) 3D geological model constructed by the HRBF algorithm (b) S1 profile built by HRBF the algorithm, (c) S2 profile built by the HRBF algorithm, (d) S3 profile built by the HRBF algorithm, and (e) S4 profile built by the HRBF algorithm**

## 4.2 Comparison of Different Algorithms

Before building the three-dimensional geological model using the SDLP and SVM algorithms, it is necessary to observe the performance of these two algorithms based on the test dataset. According to the prediction results for the test dataset, the accuracy, precision, recall, and F1 score of the SDLP algorithm are 0.982, 0.983, 0.980, and 0.982, respectively, all of which are higher than those of the SVM algorithm (Fig. 12). The reason for these overall results may be that the SDLP algorithm uses more training data, enabling the model to learn patterns with greater generalizability.

Furthermore, the accuracy, precision, recall, and F1 score of the SDLP algorithm are also greater than those of the DL algorithm (Fig.11). This phenomenon may be attributed to the increased quantity of images in the training dataset resulting from the use of pseudolabels constructed with the TIN method. The expanded training dataset enables the neural network model to achieve better generalization.

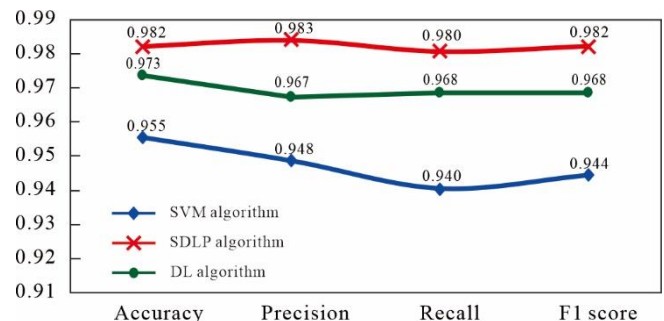

**Figure 12. Accuracy, precision, recall, and F1 score of the SDLP and SVM algorithms.**

### 4.3 Comparative analysis of models

The profiles of the 3D geological model of Shenyang city are compared to further validate the generalization ability of the
345 SDLP algorithm and the SVM algorithm. The implicit HRBF modelling method exhibits excellent consistency with the
borehole data in the profiles; thus, the profiles constructed with the HRBF method are used as a benchmark for comparison
with the profiles generated by machine learning algorithms. In Fig. 13, the horizontal axis represents the modelling results of
different algorithms for the same geological profile, and the vertical axis represents the geological profiling modelling results
of the same algorithm for different geological profiles.

In the S2 geological profile, the 3D geological models built with the HRBF algorithm and the SDLP algorithm
demonstrate a high level of consistency with the borehole data. However, the 3D geological model built with the SVM
algorithm shows relatively poor correspondence with the borehole data. Furthermore, the morphology of the formations in
the 3D geological models created with different algorithms is not entirely consistent within the S2 profile. In sedimentary
formations without fault structures, the formation boundaries typically undergo gradual changes rather than abrupt changes.
The 3D geological models generated using the SDLP algorithm or the HRBF algorithm generally adhere to these geological
laws. For instance, the intersection points of the stone-1, stone-2, and stone-3 strata and the residual-1, residual-2, and
residual-3 strata in the 3D geological models developed using the SDLP and HRBF algorithms exhibit smooth transitions,
aligning well with the sedimentation patterns of sedimentary formations. Conversely, the contact relationships among the
strata at these intersections in the 3D geological model built using the SVM algorithm do not conform to the actual
sedimentation patterns. Additionally, at the apex of the lens-shaped sand-1 formation, the 3D geological model created with
the SVM algorithm is less realistic than the models produced by the HRBF and SDLP algorithms. Guo et al. (2021)
demonstrated through 3D geological modelling methods that there are no fault structures in the Shenyang area. This finding
implies that the 3D geological model of the S2 profile built with the SVM method is not reasonable. Moreover, the HRBF
method produces modelling results that are deemed unreasonable for the lower two layers, stone-3 and residual-3, due to
365 constraints imposed by the implicit model. These constraints involve the stratum interface being defined based on the control
points of each borehole and the implicit equation. In conclusion, for the S2 profile, the SDLP algorithm exhibits the most
favourable modelling performance.

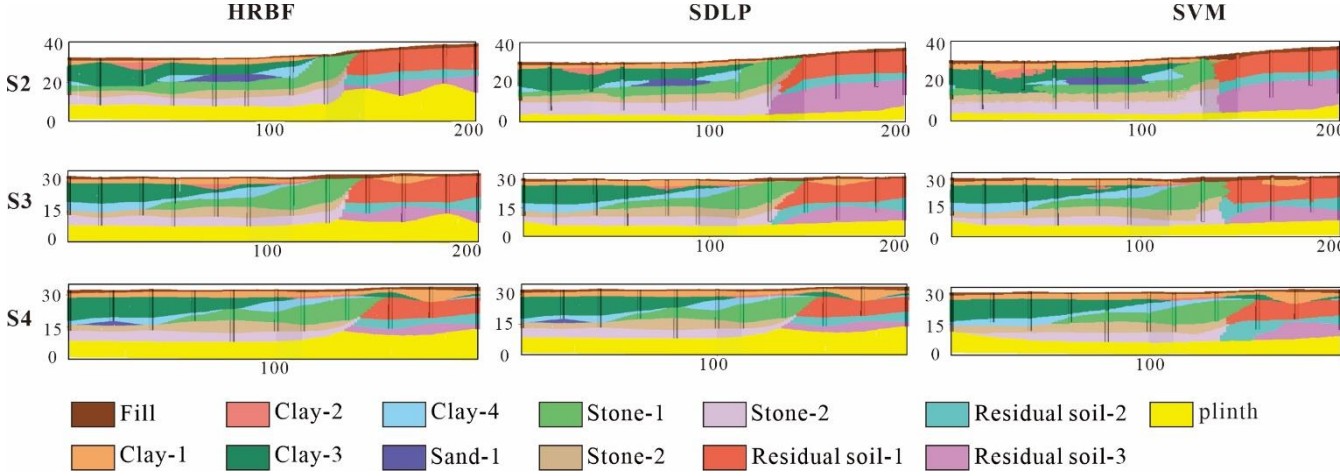

**Figure 13. Geological profiles S2, S3, and S4 for Shenyang city built based on the SDLP, SVM, and HRBF algorithms.**

The results for the S3 and S4 geological profiles are generally similar to those for the S2 profile. The 3D geological

models built using the HRBF algorithm and the SDLP algorithm demonstrate a high level of consistency with the borehole

data, and the correspondence between the 3D geological model built with the SVM algorithm and the borehole data is

comparatively poor. The boundaries of sedimentary formations in the 3D geological models built using the HRBF algorithm

or the SDLP algorithm adhere more closely to the actual sedimentation patterns than do the boundaries of the 3D geological

models built using the SVM algorithm. At the lowermost layer boundary, the 3D geological model built using the SDLP

algorithm is more reasonable than that built using the HRBF algorithm.

A comparison of the results for the S2, S3, and S4 profiles  reveal that the SDLP algorithm better reflects the borehole

data when building the 3D geological model. Additionally, the 3D geological model created using the SDLP algorithm better

aligns with the sedimentation patterns in terms of the morphology of the formations.

**4.4 Analysis of Model Uncertainty**

For a 3D geological model, only the strata boundary information reflected by borehole data is accurate, and the strata

boundaries in areas outside the borehole data region are either artificially inferred or based on constructed basis functions.

Therefore, it is necessary to analyse the strata boundaries established based on borehole data in certain areas in the three-

dimensional geological model. The implicit HRBF modelling algorithm can be used to effectively visualize borehole data.

However, because it is based on implicit basis functions for visualization, it may not effectively process the undisclosed

geological information associated with borehole data. In this study, information entropy and a confusion index are

introduced to address the inability of the HRBF algorithm to consider uncertainty in areas without borehole data. The

information entropy is calculated based on the probability distribution of all the data points in the normalized model. A

visualized information entropy model can reflect the uncertainty at different locations within the model.

In addition, the results of the information entropy and confusion index models of the SDLP and DL algorithms are compared. These results are used to demonstrate the impact of pseudolabelling on the stability of 3D geological models constructed via neural network methods.

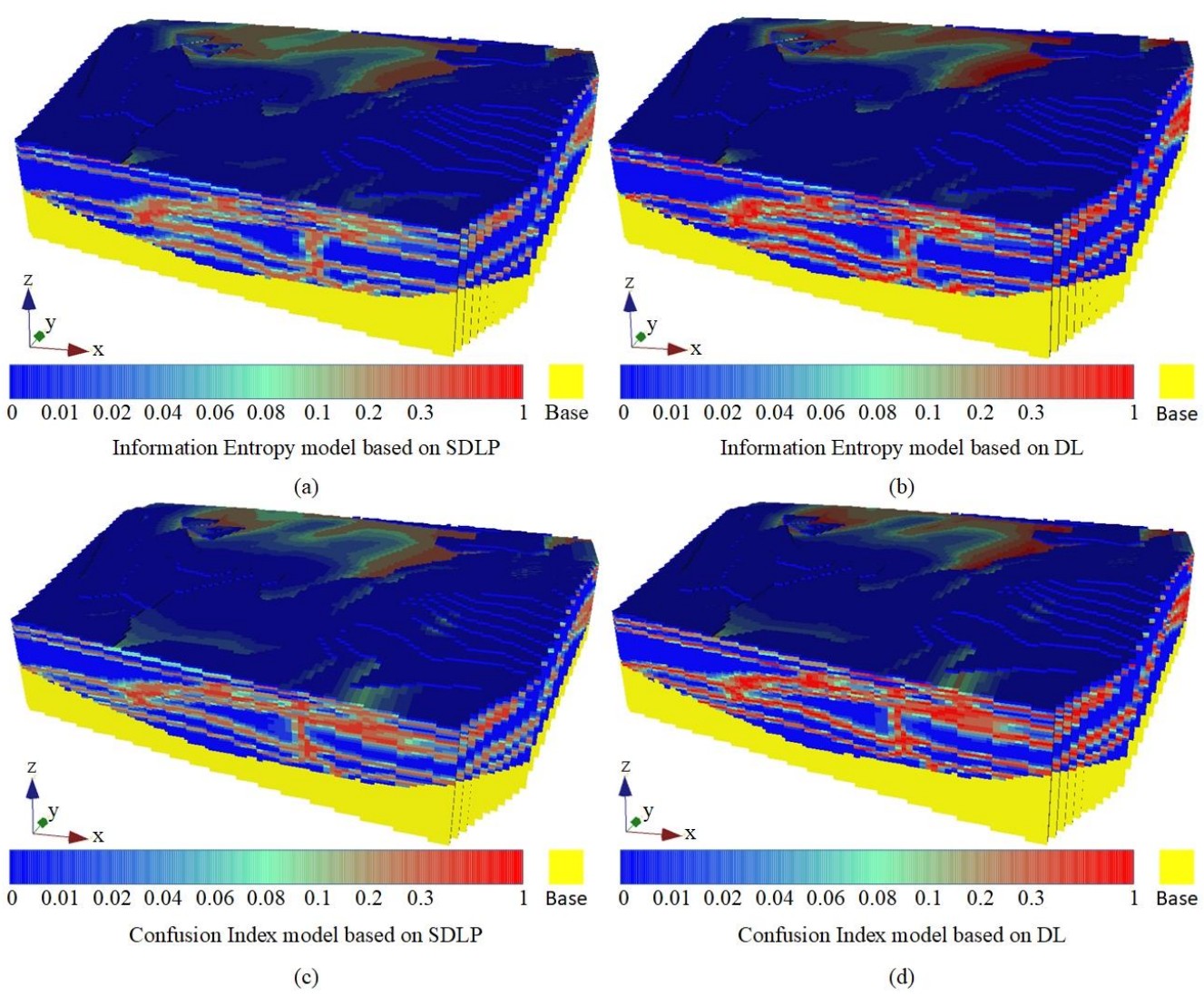

**Figure 14. Models of uncertainty: (a) information entropy model based on SDLP; (b) information entropy model based on DL; (c)**
**confusion index model based on SDLP; and (d) confusion index model based on DL**

        The information entropy and confusion index models reflect the uncertainty of the semisupervised learning method using pseudolabels and the supervised learning method used to build the models (Fig. 14). In the blue part of the information entropy model (Fig. 14a, c), where the information entropy is close to 0, the uncertainty of the stratum attribute values in the region is low, and the entropy value is small, mainly between the model stratum boundaries. In the red part, where the
information entropy is close to 1, the region has a high probability of being influenced by stratum attribute values, and the entropy value is large, mainly distributed near the stratum boundary obtained through training. In the confusion index model (Fig. 14b, d), the blue part indicates a low confusion index, and the red part indicates a high confusion index.

According to the confusion index model, the three-dimensional geological models built by the SDLP algorithm and DL algorithm both exhibit  confusion indices close to 0 within strata but increase in the confusion indices at the boundaries of the strata. The difference lies in the fact that at the strata boundaries, the confusion index of the three-dimensional geological model built with the deep learning algorithm without pseudolabelling is closer to 1, indicating lower accuracy than that of the 3D geological model built with the deep learning algorithm with pseudolabelling. Additionally, the information entropy model exhibits characteristics similar to those of the confusion index model. To visually illustrate the differences between the 3D geological models constructed by the SDLP algorithm and the DL algorithm in terms of information entropy and confusion index, the number of stable grids (with information entropy ranging from 0 to 0.01 and confusion index ranging from 0 to 0.01; Fig. 15a, b) and unstable grids (with information entropy ranging from 0.3 to 1 and confusion index ranging from 0.3 to 1; Fig. 15a, b) are recorded and compared. The results show that, compared to those of the DL algorithm, the 3D geological model constructed by the SDLP algorithm has a greater proportion of stable grids and a lower proportion of unstable grids. The findings demonstrate that utilizing the TIN algorithm to construct pseudolabels can enhance the stability of the model.

The information entropy and confusion index models can be used to overcome the inability of the HRBF algorithm to consider uncertainty, and the results demonstrate that the SDLP algorithm is superior to the deep learning algorithm without pseudolabelling for constructing 3D geological models from the perspectives of information entropy and the confusion index.

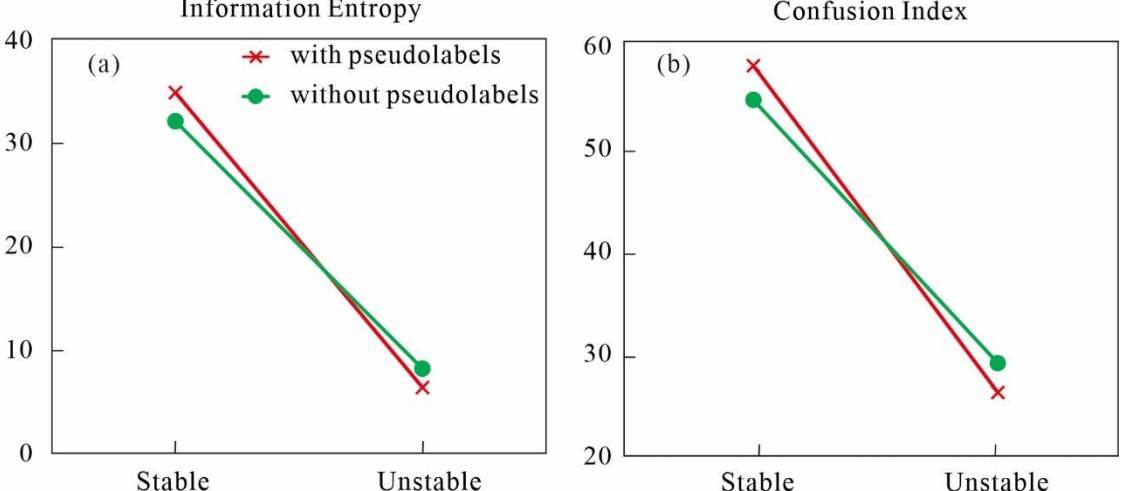

**Figure 15. Line plot of the information entropy(a) and confusion index (b).**

## 5. Conclusion

In this study, we propose semisupervised deep learning using a pseudolabelling algorithm to construct a 3D geological model based on borehole data. By labelling the grid data with high accuracy using the explicit TIN modelling method, we address the lack of labelled training data for building deep learning models. The original data for this study were obtained from an engineering borehole dataset from Shenyang city, and 3D geological models of Shenyang city were constructed

using the SDLP, SVM, and HRBF algorithms. On the test dataset, the SDLP algorithm outperforms the classical SVM machine learning algorithm, with an accuracy, precision, recall, and F1 score of 98.16%, 98.3%, 98.0%, and 98.2%, respectively. Moreover, the 3D geological model constructed using the SDLP algorithm accurately reflects the boundaries of the formations in the borehole data and aligns well with the real sedimentation patterns. The 3D geological models constructed by the SDLP algorithm overcome the inability of the implicit HRBF modelling algorithm to consider uncertainty. In conclusion, the proposed SDLP algorithm provides a solution for the lack of training data in deep learning and fills the gap that cannot perform uncertainty analysis of the HRBF implicit modelling method.

*Code availability*. The PDNN was written in the Python programming language. The program reads borehole data and preprocesses the borehole data with upsampling and normalization. By using the DNN to train the model and predict the attributes of the data points, pseudolabels with high confidence scores were added to the unlabelled grid points. The code is available for download from the following public repository: https://zenodo.org/deposit/7833570.

*Data availability*. The model data and terrain data used in the case study in this paper are available at: https://doi.org/10.5281/zenodo.7535214.

*Video supplement*. We have provided web links to download the video recordings of our case studies. A case study of a real area verifies the feasibility of the proposed approach. The video supplement can be viewed at: https://drive.google.com/file/d/13VERDXM6YJmP7xMabQy3IjhCExuQSWzk/view?usp=sharing.

*Author contributions*. Xuechuang Xu and Jiateng Guo conceived the manuscript; Jiateng Guo provided funding support and ideas; Xuechuang Xu was responsible for the research methods and program development; Jiateng Guo provided the data used in this research; Xuechuang Xu, Jiateng Guo, Xulei Wang, Lixin Wu, Mark Jessell, Zhibin Liu and Yufei Zheng helped improve the manuscript; and Luyuan Wang helped modify the manuscript. All the authors have read and agreed to the published version of the manuscript.

*Competing interests*. The authors declare that they have no known competing financial interests or personal relationships that could appear to have influenced the work reported in this paper.

*Acknowledgements*. This work was financially supported by the National Natural Science Foundation of China [grant number: 42172327], the State Key Laboratory of Disaster Prevention & Mitigation of Explosion & Impact [grant number: LGD-SKL-202209], and the Fundamental Research Funds for the Central Universities [grant number: N2201022].

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
