# Peer review of "A Semisupervised Deep Learning Neural Network Using Pseudolabels for Three-Dimensional Shallow Strata Modelling and Uncertainty Analysis in Urban Areas from Borehole Data"

_Geoscientific Model Development, 2023_

## Author Response (AR1)

Dear Editor and reviewers,

we would like to thank you for your kind letter and for reviewers' constructive comments concerning our manuscript (**A Semisupervised Deep Learning Neural Network Using Pseudolabels for Three-Dimensional Shallow Strata Modelling and Uncertainty Analysis in Urban Areas from Borehole Data**). These comments are all valuable and helpful for improving our article. All the authors have seriously discussed about all these comments. According to the reviewers' comments, we have tried best to modify our manuscript to meet with the requirements of your journal. Some main modifications as follows:

(1) We have rewritten the introduction section based on the comments from the reviewers. In the new introduction, we emphasize the lack of training data and the issue of imbalanced training data when using deep learning methods to construct three-dimensional geological models, which has received relatively less attention in current research. Furthermore, our proposed method also addresses the problem of deterministic implicit modeling that cannot perform uncertainty analysis. Through these descriptions, we aim to address the reviewer's concern regarding the lack of clarity regarding the innovative aspects of our work.

(2) We have revised the discussion section based on the reviewer's comments. In the new discussion, we provide a horizontal comparison of the advantages and disadvantages of the three modeling methods mentioned in this paper. Additionally, we provide a vertical comparison of the impact of adding pseudo-labels to the model. Through these revisions, we aim to address the reviewer's concerns regarding the unclear relationships between the three algorithms and the unclear role of the TIN algorithm.

(3) In the experimental section of the manuscript, we have modified our code based on the reviewer's comments to output the accuracy, precision, recall, and F1 score of different algorithms on the test dataset. Through the comparison of these data results, we aim to intuitively select the optimal algorithm among the three mentioned in this paper. Additionally, we have also modified other sections of the code, such as re-generating profiles of the 3D geological model, information entropy, and confusion

indices.

(4) In the figures section of the manuscript, we have carefully revised our illustrations based on the reviewer's comments.

(5) We have also provided a re-description of the fuzzy concepts such as "geological semantics" in the manuscript.

**Detailed modifications and responses are as follows:**

**Reviewer #1**

**Specific comments:**

Line 14: I'm not sure what "urban geology exploration" refers to.

"Urban geology exploration" refers to "urban geological surveys". We have modified the unclear words.

Line 16: Why would we need more complex models? What we want is to support decision making, and more complexity doesn't necessarily means better support (on the contrary).

We want to solve more complex problems with more complex models. For example, our study region in the paper contains many stratigraphic truncations and lenticular bodies, and these tectonic features cannot be resolved by simple models.

Line 16: "analysing the modelling results with uncertainty" I'm not sure about the phrasing there, maybe replace "with" by "and their"?

We have modified "analysing the modelling results with uncertainty", and we stated specific reasons.

Line 17: "built" instead of "establish"?

Thank you for your suggestion. We have modified this word in the manuscript.

Line 18: What makes this survey complex? The number of data? A fine stratigraphic layering? A folded and faulted structural setting?

The reason for the survey complexity is that there are many irregular layers and features, such as folded and faulted layers. In addition, we have added additional

descriptions to the manuscript.

Line 19: What is a traditional machine learning method? How is it different from the method proposed in the paper?

Traditional machine learning methods refer to algorithms and techniques that were developed earlier and are widely applied in the field of machine learning. Examples include support vector machines (SVMs) and random forests. These methods are suitable for small-scale datasets and perform well for simple and structured tasks. In contrast, the deep learning algorithm employed in our research utilizes a multilayer neural network structure and learns features and patterns of the data through forward and backward propagation algorithms. The approach we adopt is more effective in dealing with large-scale datasets and complex tasks.

Lines 19-20: How is the uncertainty analysis performed?

Response: The final layer of the neural network utilizes the softmax function to convert the output into probability values corresponding to different classes. By computing the information entropy and confusion indices based on the predicted probability vectors for each grid cell in the 3D geological model, an uncertainty analysis of the model can be performed.

Line 20: What does "expanding the sample space" mean?

Response: "Expanding the sample space" refers to the method of expanding the training dataset by generating pseudolabels for unlabelled data. Our intention is to solve the problem of supervised learning requiring many labels.

Line 21: What does "geological semantics" mean?

"Geological semantic" means the relationship shown in the borehole data. We have removed the ambiguous statement.

Line 22: More complex regions than what region? The one from the case study? Is that something that was tested? If not, how do you know?

We apologize that our careless presentation led to confusion. We intended to convey that it is a complex geologic area similar to the one modelled in this paper. We have removed this text.

Thank you for your careful guidance on the abstract. We have reworked the abstract based on your suggestions.

Line 24: What do you mean by "Geological spatial distribution" exactly? Distribution of facies? Of rock properties?

"Geological spatial distribution" means "The distribution of geological tectonics and strata in urban underground space". We have removed the ambiguous statement.

Line 25: What do you mean by "underground situation"? What are the properties of interest? How far deep do we need them?

"Underground situation" also means "The distribution of geological tectonics and strata in urban underground space". We have removed the ambiguous statement.

Line 26: How to determine whether a geological model is reasonable or not?

The term 'reasonable geology' means that the constructed model effectively reflects the actual underground conditions. We have removed the ambiguous statement.

Line 26: What does "intuitive expression of geological features" mean?

The phrase "intuitive expression of geological features" means that we can directly reflect the borehole data with a 3D geological model. We have removed the ambiguous statement.

Line 27: "revelation of the spatial distribution law" That should be part of the conceptual model used as basis to build the geological model (e.g., do we have a channelized sedimentary environment or a carbonate platform), not really of the geological model itself.

"Revelation of the spatial distribution law" is related to the relationships among different strata. We have removed the ambiguous statement.

Line 28: "a long geological process" Usually it is the results of multiple processes (e.g., water flow and sediment transport in rivers, wave action, diagenesis, folding).

We want to use the description of "a long geological process" to express the complexity of the study area. We have removed the ambiguous statement.

Line 29: We do have those laws, and they are implemented with various degrees of approximation in forward stratigraphic models in the case of sediment transport for instance. But using those models remain costly.

You are correct. This was an error in the description, and we have removed this text.

Line 31: "powerful computing power of computers" The phrasing is weird here, and this is true of any numerical method, not just deep learning.

You are correct. This was an error in our description, and we have removed this text.

Line 31: "complex fields" Which ones? What makes them complex?

"Complex fields" refers to fields such as mineral exploration, petrographic identification, and palaeobiological recognition. We have modified this description.

Line 31: "increasingly attracted the attention of geological researchers, such as 3D modelling" Any reference?

We have added references.

Line 36: What does "intuitively" mean here?

"Intuitively" means that it is easy to obtain the properties and thicknesses of geological layers, whether through visually observing boreholes or examining databases. We have modified this description.

Line 37: The distinction between explicit and implicit modeling was defined long before Wang et al. (2018).

Thank you for your suggestion. We have added references.

Line 37: What is a geological semantic constraint?

"Geological semantic" means the relationships shown in the borehole data. We have removed the ambiguous statement.

Line 38: Which geological laws?

"Geological laws" refer to the unconformities of strata, sequential tectonic influences, etc. We have removed the ambiguous statement.

Line 42: Which implicit equations? Your explanation of implicit modeling is a bit confusing, and implicit modeling is not always based on basis functions.

Thank you for your suggestions. "Implicit modelling based on basis functions" is indeed too narrow; we have modified this description.

Line 52: "has been developed as a method for boreholes" What does that mean?

This was a mistake, and we have removed the sentence.

Lines 48-54: The link between stochastic simulations and implicit modeling is unclear. And Lancaster & Bras (2002) is not a stochastic simulation method.

Thank you for your advice. There is a lack of clarity regarding the connection between stochastic simulation and implicit modelling, and there was an erroneous reference made to the stochastic simulation method.

Line 60: What does "mined data" mean?

"Mined data" refers to the ability to learn features from raw data. The text was changed to "mining data".

Line 64: What does it mean to "intelligently generate" a model? This sounds like an unsupported value judgment.

"Intelligently generate" refers to the ability to use deep learning techniques to process multisource data and generate regional three-dimensional geological models. There may be an issue of inappropriate wording in this context. We have modified the sentence.

Line 67: "has been realized" This sounds like a long list of papers using deep learning just for the sake of making a list. What's their relation to this study?

In our rewritten introduction, we presented examples of machine learning combined with geological analysis and summarized the problems when applied to 3D geological modelling. Deep learning facilitates us in solving the above problems.

Line 72: There are a lot of repetitions all along the introduction, which would benefit from a reorganization to be clearer, better introduce the context and problem statement, and better link to previous studies actually related to this work.

Thank you for your suggestion. We have reorganized the introduction.

Line 75: How do those two approaches relate to all the methods mentioned before? How do they relate to explicit and implicit modeling.

These two approaches involve the integration of deep learning with urban borehole data for three-dimensional geological modelling. Like the previous methods, they are also techniques for constructing three-dimensional geological models but further restrict the choice of data sources. In comparison to explicit and implicit modelling,

these are distinct modelling methodologies.

Line 91: There needs to be a stronger and more explicit link to all the studies mentioned before. How does the previous paragraph relates to this study is unclear.

We have rewritten this section and provided a stronger and more explicit link.

Line 93: "the pseudolabel data with high confidence" What are those data? Where do they come from? This is either too little or too much detail at this stage, and I'm quite confused about the objectives of the paper.

We have rewritten this section and provided information on how to use pseudolabel data.

Thank you for your suggestions regarding the introduction section of our paper. Based on the feedback from the two reviewers, we have rewritten the introduction. In the new introduction, we first emphasize the importance of three-dimensional geological modelling and carefully describe the classification of borehole data for 3D geological modelling, as well as the issues associated with it. Building upon these problems, we introduce machine learning methods and further highlight the problems encountered when using machine learning to construct 3D geological models, leading to the introduction of our semisupervised learning approach based on pseudolabels.

Line 99: This is true all the time, not just with deep learning.

Response: Here, "the problem of classifying borehole data" refers to the prediction of geological categories. The explicit or implicit modelling of geological "classification" does not fall within the scope of prediction. Furthermore, as mentioned in the introduction, the simulation of borehole sequences includes both sequential stratigraphy prediction and thickness prediction, which involve not only category prediction but also the prediction of geological layer thickness. These factors present distinct conceptual differences.

Lines 101-102: "the model at the borehole should be as consistent as possible with the stratum information revealed by the current borehole" I'm not sure what that means, is this about upscaling?

Response: This sentence means that the geological formations revealed by the

borehole data are accurate. Therefore, it is necessary to train a neural network based on the geological information provided by the borehole data to build a realistic 3D geological model that conforms to the borehole data. To avoid ambiguity, this sentence has been removed from the manuscript.

Line 104: "To increase the amount of data, the borehole data are upsampled" why is that needed?

Response: Because the original borehole data are continuous one-dimensional data, they need to be discretized before being input into the neural network. Therefore, the borehole data need to be upsampled and converted into discrete point data. By upsampling the borehole data, we can obtain more geological samples, thereby better capturing the variations, features, and continuity of the geological formations.

Figure 1: I do not understand what is going on here, and the text does not help. What do the sections mean?

Response: Lines 124-135 provide a textual explanation of the content in Figure 1. This section explains the connectivity of the profiles in Figure 1 using the tin network. Each triangular network consists of three boreholes, and if the geological formations between any two boreholes have the same attributes, they are connected to form a deterministic profile. After connecting the deterministic profiles, they are horizontally sampled at regular intervals based on modelling accuracy and vertically sampled at unequal intervals. We have added labels to Figure 1 and referenced the figure name in the corresponding text.

Line 124: "direct or indirect" Why "or"? What is their indirect role?

Response: Borehole data provide direct observational results of actual geological information, which can be used to validate or calibrate the accuracy of geological models. This is a direct effect. The interconnected profiles between boreholes, to some extent, reveal the trends and dips of geological formations, which is an indirect effect. The phrase "direct or indirect" is used to emphasize the dual impact of borehole data on model generation. Here, "or" should be replaced with "and" for clarity.

Line 124: What is "geological semantic information"? And what is "geological semantic information with high reliability"? How do you assess whether geological information is reliable or not?

Response: "Geological semantic information" here refers to the direct observational results of actual geological information provided by borehole data, as well as the interconnected profiles between boreholes, which to some extent reveal the trends and dips of geological formations. Since the model is built solely based on urban borehole data, which are provided by professionals, as researchers of the modelling algorithm, we consider the geological information obtained from the borehole data to be reliable unless there are obvious errors.

Line 126: What is the Delaunay rule?

Response: The Delaunay rule is the basis of Delaunay triangulation. Delaunay triangulation refers to constructing a triangular mesh given a set of points, such that no point lies inside the circumcircle of any triangle in the mesh. In Delaunay triangulation, the circumcircle of each triangle cannot contain any other points, ensuring the quality and stability of the mesh. Delaunay triangulation is commonly applied in the field of geological modelling.

Line 126: I'm really confused about the reason we need all this.

Response: Delaunay triangulation can be used to optimize mesh quality and generate high-quality triangles. A triangular mesh can be automatically created based on the density and distribution of borehole data, thereby effectively reflecting the variations in geological structures. This approach ensures that all vertices within each triangle are nearest neighbours. Utilizing the information from adjacent data points improves the accuracy and continuity of the generalized prisms for the lithological attributes obtained from surrounding borehole data.

Line 130: What is a "GTP-like section connection method"?

Response: The generalized tri-prism (GTP) method is a classic explicit modelling approach used to constructs a TIN (triangular irregular network) during the modelling process and considers the lithological attributes of tetrahedral prisms based on the

borehole data forming those prisms. This process involves connecting stratigraphic profiles between boreholes. In the text, the term "GTP-like section connection method" is used to describe the process of connecting stratigraphic profiles between boreholes, where only sections with identical lithological attributes are connected.

For detailed information on the GTP method, please refer to the following article:

Wu, L.X. : Topological relations embodied in a generalized tri-prism (GTP) model for a 3D geoscience modelling system, Computers & Geosciences, 30(4): 405-418, https://doi.org/10.1016/j.cageo.2003.06.005, 2004.

Line 131: "can simulate a variety of complex geological phenomena" What does that mean? How can a TIN simulate anything?

Response: There may be a misunderstanding here. It is not stated that TIN can simulate anything, but rather that the GTP method can model complex geological situations such as geological bifurcation, thinning, and faults.

Line 132: "modelling scope of this study is mainly for a quaternary sedimentary surface" That needs to be specified in the introduction.

Response: In the manuscript, we clarify that the "modelling scope of this study is primarily focused on a Quaternary sedimentary surface".

Line 133: "strata are deposited in chronological order" Geological layers are always deposited in chronological order.

Response: Yes, here, we are just trying to convey that the stratigraphy has not been affected by other geological phenomena.

Line 136: But by normalizing like that (x,y) and z don't follow the same scale anymore. How does that affect the results?

Response: After normalizing x, y, and z, they are mapped to the range of [0-1]. In this case, during the training process, different features have equal influences on model training, ensuring convergence. Further explanations of this method can be found in the reference listed in the next response.

Line 144: I cannot find any proof of that.

Response: The reference provides an explanation of the benefits and principles of normalization. In the experimental process, normalization has proven to be beneficial

for the training of networks.

Orr, G. B., & Klaus-Robert Müller. (1998). Neural networks: tricks of the trade. Lecture Notes in Computer Science. https://doi.org/10.1007/3-540-49430-8.

Line 146: Huang et al. (2012) didn't define the single-layer perceptron, this was done in the 40s and implemented in the late 50s.

Response: We acknowledge that certain references in the manuscript do not cite the original research, and appropriate changes will be made in the subsequent revisions.

Line 148: Activation functions are more important to capture non-linearity, and a single layer neural network is a universal approximator. So multiple layers are not always needed.

Response: Indeed, activation functions are important for capturing nonlinearity, and a single-layer neural network is a universal approximator. However, their effectiveness in solving complex nonlinear problems is limited. Here, we want to express that multilayer perceptrons (MLPs) are better suited for tackling complex nonlinear classification problems.

Line 152: "input index and output index" What are those indices? So it doesn't use the actual values?

Response: Actual values are used. However, for certain attributes, such as the categories of geological formations, numerical representations such as 1, 2, 3, and 4 are used. These are simply expressions of the input and output of the model.

Line 152: "which is a multilayer feedforward neural network" No the result is a prediction, the neural network is a model to get that prediction.

Response: This sentence does indeed have a grammatical issue. The entire sentence was corrected. The sentence in the original text was correspondingly modified.

Line 156: The phrasing is confusing, there is a weight associated to each neuron.

Response: The sentence is not accurately expressed and should be replaced with " Input data are output after passing through multiple hidden layers and compared with the expected label to obtain the corresponding error, which is then propagated layer by layer backwards to adjust the weight of each layer". The sentence in the original text was correspondingly modified.

Line 158: "The data in the data set are output after the multilayer perceptron" What does that mean?

Response: The sentence is not accurately expressed and should be replaced with "The data from the dataset is passed through the multilayer perceptron, and the outputs are compared with the expected values".

Figure 2: It should be "prediction", not "pridiction".

Response: We apologize for the incorrect spelling. The term was corrected in the revised manuscript.

Line 176: "borehole data tend to be dispersed" What does that mean? Aren't your data point cloud data in that case?

Response: Point cloud data and unequal interval sampled borehole data can display significant differences, primarily in terms of data format and data density. Point cloud data are composed of a large collection of discrete points, each containing coordinate information and possibly other attributes. Point cloud data can have an irregular distribution and varying density because they are generated based on sampled points from real-world scenes or objects, whereas unequal-interval sampled borehole data are selected based on specific rules or algorithms. Thus, there are still significant differences.

Line 179: But that upsampling can lead to imbalance and biases.

Response: Because these are the only reliable data that can be obtained, sampling for deterministic profiles may result in imbalance and bias, but information loss is avoided.

Line 184: "no specific mathematical law for the attribute of strata" I don't understand what that means.

Response: Due to the diversity and nonlinear nature of geological processes, geological attributes cannot be simply described by mathematical rules. Therefore, we use a triangulated irregular network (TIN) to ensure that all vertices within each triangle are nearest neighbours, thus ensuring the accuracy and continuity of the

prisms of geological attributes obtained based on surrounding borehole data.

Line 189: Which model is used here? It's not clear to me what is the advantage of doing that, why not using that model directly everywhere?

Response: The model used here is trained with only labelled data, and the purpose is to predict pseudolabels for unlabelled data. By using this model, the unlabelled data can be effectively utilized, and the performance of the trained model is better than that of a model trained solely on labelled data.

Line 222: "SVM method" Is that the HRBF mentioned above? What's the difference?

Response: The SVM method is not the same as the HRBF method. The support vector machine (SVM) is a machine learning method, while the Hermite radial basis function (HRBF) is an implicit surface construction method.

Lines 224-225: "The model established using the algorithm mentioned in the experiment is visualized with the developed visualization platform" I don't understand what that means.

Response: The sentence was not accurately expressed. We have removed the ambiguous statement.

Line 218: What is the "rationality" of a model?

We apologize for not describing the "rationality" in the text. We have removed this vague description.

Line 221: At this stage there are still no explanation of what "geological semantics" means.

"Geological semantic" refers to the relationships encompassed in the borehole data. We have removed the ambiguous statement.

Line 235: That is a very high sampling rate (and unusually regular, but that might be because it is related to a geotechnical study?), do we really need advanced modeling methods there? There is no mention of why a 3D model is needed, so it is difficult to judge that here.

Our work is based on urban area borehole data. These data are mainly high-density borehole data in shallow layers of urban areas and are primarily obtained through urban geological surveys. Due to restrictions imposed by buildings, underground

infrastructure, and pipelines in urban areas, the boreholes used for urban geological surveys are generally shallow. Owing to the complexity of geological conditions in urban regions, as well as the needs of urban infrastructure development and land utilization, urban geological survey boreholes often require a high-density distribution within a given small area. The objective application of urban geological modelling in this study is to support engineering design and construction, land planning, and development. The goal of constructing a three-dimensional geological model for urban areas is to provide comprehensive geological information, such as the distribution of underground soil layers and the spatial distribution of different lithologies. Through the three-dimensional model, a better understanding of underground geological structures can be achieved, providing a scientific basis for decision-making in related fields. Previous methods of model construction have generally been deterministic. In contrast, our proposed method not only yields a higher-precision model but also provides information about model uncertainty, offering a reasonable basis for decision-making.

Line 245: I do not think "confused" is the right term here. "Missed"? Or "mislabelled"?

This sentence uses inaccurate terminology. "Confused" should be replaced with "incorrect prediction". The sentence in the original text was correspondingly modified.

Line 245: Recall, precision, and F1 would be more adequate metrics than accuracy, especially if the layers have variable thicknesses.

Thank you for the suggestion. We have added precision, recall and F1 score results as well as relevant discussion.

Line 246: What does "depositional termination" mean?

This was a lack of precision in our wording, and we have removed this vague expression.

Figure 5: Are the labels the different layers? It's unclear what "label" means here.

Yes, the labels represent different geological strata.

Figure 6: Same comment here: are the classes the different layers? And what are the micro- and macro-average ROC curves? Different averages for all the layers? It would be better to have them above the other curves, they are difficult to see.

Yes, class represents different geological strata. Microaverage ROC curves use the overall true positive rate (TPR) and false positive rate (FPR) as points on the microaverage ROC curve. Macroaverage ROC curves average the TPR and FPR across all classes to obtain the macroaverage TPR and FPR and use them as points on the macroaverage ROC curve. The microaverage ROC curve emphasizes overall performance, while the macroaverage ROC curve focuses on the independent performance of each class.

Figure 6: There seems to be large differences between the different layers, why is that?

The main reasons for the differences are primarily due to imbalanced sample quantities between different geological strata, uneven sample distributions, and variations in the classification performance of the models across different geological strata.

Figure 6: The use of the same color for multiple classes makes it difficult to analyze the plot, but are the areas right? Class 11 is said to have an area of 1, yet on the plot its true positive rate doesn't go straight to 1, so there seems to be a big issue there.

There was indeed an issue with duplicate category colours here, and we have modified the image. The ROC curves were calculated using the sklearn library. Category 11 is reported to have an area of 1, but its true positivity rate on the graph does not reach exactly 1. This discrepancy may be due to rounding errors in the calculations. We have expanded the number of retained valid decimals in the new image.

Line 259: It's still unclear to me what is the use of the TIN mesh.

The function of a TIN was explained in the previous question.

Line 266: Is the cross-validation based on the borehole data or on the boreholes? This is important, because spatial correlations mean that the former will underestimate the prediction error.

The cross-validation was based on the borehole data, as now shown in Figure 10.

Line 278: "three depositional terminations between any stratum and the surrounding boreholes" I don't understand what this means.

Our intention was to avoid removing boreholes that contain lenticular bodies or

contain information about stratigraphic boundaries during K-fold validation.

Lines 276-278: But by doing so you're creating a bias in your validation. In a real application you will miss locations that would be critical to understand the layers' distribution, and that is what needs to be tested to have a robust validation of the method.

Yes, this approach may lead to bias, so we provided the results of a robust validation of the method in Lines 273-274.

Line 286: "excluded sample K1 test borehole data" Which one is that?

The "sample K1 test borehole data" include data from boreholes "20, 22, 39, 52, 62, 74, 81, 83, 94, 101, 117, 121, 128, 130, 132, 142, and 154", which were randomly selected during the experimental process. We have added this detail to the paper.

Line 290: What happens if you remove more than one borehole though?

In this case, the prediction accuracy would continue to decrease.

Figure 10: What about the uncertainty here?

The uncertainty of the model would increase compared to that before removing the boreholes.

Line 296: What is the "rationality" of a model? And what is a "mature" modeling method?

This was a lack of precision in our wording, and we have removed this vague expression.

Line 297: Any reference for this method?

We have added some references in the new discussion.

Line 299: I'm not sure I understand what a vector model is. The implicit model can predict a value for the scalar field at any location, so why is there a need to transformed the predictions into a grid?

Here, we describe this method to facilitate the next step of comparative analysis.

Line 303: There are so many data available that this is hardly surprising.

What we are trying to do here is to prepare us to supplement sparse large-scale data with rich small-scale data.

Line 310: "predicts [...] with high confidence, which has certain uncertainty" How can prediction have a high confidence yet have a certain uncertainty?

This was a lack of precision in our wording, and we have removed this vague expression.

Line 321: "modelling results of the proposed method for complex geological conditions are significantly improved compared with those of the SVM method" This is a judgment call without actual proof. Why not apply the cross-validation to the HRBF and SVM models as well? This would make the results much stronger.

Thank you for your suggestion. We have described this issue in detail in the new Discussion section. Regarding your suggestion to cross-validate HRBF and SVM, it is a very good suggestion. However, due to time constraints, this process will be a focus of our next study.

Line 323: The consistency of which feature? This is a very subjective and partial validation.

This was a lack of precision in our wording, and we have removed this vague expression.

Line 325: "consistent with the geological semantics" What does that mean? How do you check that in an objective way?

This was a lack of precision in our wording, and we have removed this vague expression.

Lines 327-328: You cannot conclude that from a single section in a single case study, this is too high a jump without quantified justification.

Thank you for your suggestion. It is true that we were inaccurate in our description here, and we have changed our incorrect description.

Figure 13: The colorscales don't make any sense, 0.3 on the left side looks like 0.4 on the right side.

Because the colour proportions on the left and right sides are different, it is normal for 0.3 on the left side and 0.4 on the right side to represent the same colour situation.

Figure 13: There seems to be an issue with the labeling of the sub-figures, top and bottom are the same.

This was an error caused by our carelessness, and we have fixed the error.

Line 349: What's a significant decrease? Based on the plots, the difference appears quite limited, and its really not clear that the semi-supervised method outperforms the supervised method. Also the validation here remains qualitative (comparing plots visually), it would be better to have summary statistics to compare the two methods in a quantitative way.

Thank you for the suggestion. We have revised the graphic and have come to the same conclusion.

Line 359: I'm not sure I understand the problem actually. From boreholes we get the horizons, so can directly interpolate between horizons, or sample data points within layers, which sounds like what is done here, but I would hardly qualify that as novel. And in the second case we loose the exact location of the horizons (unless one samples very finely), and looking back at the method section, it is not clear to me how this is tackled.

We have updated the innovative points of this article. We have reworked the entire discussion.

Line 367: I don't understand the comparison to MPS specifically. And a grid is used here too so I don't understand the point you're trying to make here.

Thank you for your suggestion. We have removed this part of the description.Line 383: I think you mean "domain" instead of "scope".

Thank you for your advice. We have changed this term.

Line 386: "for sampling data points" What do you mean by that? A randomized K-fold cross-validation? Was this mentioned before?

We have modified these descriptions.

Line 392: "has more advantages in dealing with more complex geological phenomena" This is not supported by the results.

Thank you for your suggestion. It is true that we made a mistake in the description here, and we have refreshed the innovative points of this article. We have reworked this part of the description.

**Reviewer #2**

- 24 "reasonably" for which criteria?

We apologise that our vague descriptions have caused you distress. We have removed the ambiguous statement.

- 26 again using subjective words like "reasonable" and "intuitive"; please try to be more

specific.

We apologise that our vague descriptions have caused you distress. We have removed the ambiguous statement.

• 28 what is the purpose of mentioning the temporal distribution when there is no notion of time in the rest of the paper?

We apologise that our vague descriptions have caused you distress. We have removed the ambiguous statement.

• 29: mechanical laws are rather well understood and can capture the stratum distribution well. The challenge is in applying them (with the appropriate distributions of material properties and boundary conditions).

You are correct. This was an error in our description, and we have removed this section of text.

• 52: more information is needed to understand "defining a random simulation path". Which simulation? Why on a path? Why a random path?… Or, why mention that anyway? This is repeated in the text, so it needs to be really clear.

We apologise that our vague descriptions have caused you distress. We have removed the ambiguous statement.

• 74: what other data than "spatial data" come from boreholes?

The original borehole data mainly included borehole coordinates (X, Y), elevation, lithological thickness, lithological bottom depth, borehole number, and lithological ID number. We have provided these details in section 2.

• 79: The lithology simulation is a spatial point simulation I guess, please rephrase

Thank you for your suggestion. We have modified the corresponding text.

• 82: mentioning B-splines (2D curves) and (3D) voxels. Do you mean NURBS?

In fact, this is not what this article is addressing and does nothing to advance the article; we have removed this section.

• 82 what is the criterion for those curves? I guess separation/clustering problem but it needs to be mentioned explicitly

In fact, this is not what this article is addressing and does nothing to advance the article; we have removed this section.

• 83: why is the model more accurate?

In fact, this is not what this article is addressing and does nothing to advance the article; we have removed this section.

• 84: how is the randomly selection of the curves reconciled with its automatic selection (l.82). In the absence of precision l.82, I had imagined the simplest option, which is

deterministic. Why random?

In fact, this is not what this article is addressing and does nothing to advance the article; we have removed this section.

- 85: what do you mean by "overthrow the order of strata"? Why is it lower in accuracy?

Here, we express the possible difficulties in building 3D geologic models of complex strata using the implicit equation approach. However, since our representation was unclear, we have rewritten this section.

- 91: the link between that last paragraph of the introduction and the previous conclusion about lithology prediction is not explicit enough.

Based on your suggestion, we have rewritten the introduction section.

- 100: assuming vertical boreholes? How does the method work for arbitrary shapes of boreholes?

This study focuses on a vertical borehole. If the method proposed in this paper is applied to cases of slightly inclined borehole trajectories with no crossing, it still has certain applicability. However, for horizontal boreholes or high-angle deviated boreholes, more data preprocessing and model adjustments may be needed.

- 115: Hij doesn't appear in eq.1.

'Hij in the figure indicates the jth sample in the ith layer of a certain borehole' has been removed from the text and added to the explanation of Figure 1.

- 130 "GTP" acronym undefined (I guess from Wu 2004)

In the new manuscript, we have added the definition of the acronym 'GTP'.

- 167: rephrase "multiplying the weight matrix"

In the manuscript, we rephrased this sentence. "Each input represents a spatial feature dimension, and through four fully connected layers, the input data are processed and transformed. Each hidden layer contains multiple nodes, where each node is connected to all nodes in the previous layer. By multiplying by weights and applying an activation function, the input undergoes nonlinear transformation, resulting in expanded dimensionality."

- 169: what are 'high-dimensional features"?

Through the forward propagation process of the neural network, the input low-dimensional data undergo a series of linear and nonlinear transformations, gradually transforming into high-dimensional features. The specific meanings of these high-dimensional features are often learned automatically by the neural network rather than

being manually defined. In the high-dimensional space, the neural network can better learn the relationships inherent in the data, thereby improving the model's classification ability.

- 2 "pridiction" typo

The incorrect spelling used in the figure has been corrected in the new manuscript.

- 179-180: rephrase that (unreadable) sentence ("Only by ⋯ limited features")

In the manuscript, we rephrased this text: "Supervised learning depends on a large quantity of labelled data to enhance model performance. The labelled data used for training 3D geological models are obtained by upsampling limited borehole points and deterministic borehole profiles. Labelled data associated with spatial grid points in urban areas, which require high modelling precision, are scarce and contain very few features".

- 187: why the highest confidence and where ("in this range")?

Here, a deterministic modelling method called the GTP method is referenced. It can be used to determine the formation properties within a prism based on the connections between borehole strata, and it provides high confidence regarding the formation attributes within the prism. Of course, it is not assumed that this method, or any single method, is necessarily completely correct, but it is reliable enough to serve as a reference. When making predictions for unlabelled data, if the predicted result is consistent with the formation properties of the prism where the data point is located, the corresponding pseudolabel is added.

- 222: definition of mu_max: the probability of ⋯

The incorrect expression used here has been appropriately revised in the manuscript.

- What is the purpose of section 3.1? (What is the relevance of the weather information?)

We have removed this section.

- 237: how is the training accuracy evaluated? How much data is used for calibration vs testing?

We have added the precision, recall, and F1 score for the test dataset.

- 4 The loss should be shown in log scale. (There's also a typo "accurary")

We have modified the figure.

- 243 loss function is "close to zero", not "poor"

Thank you for your suggestion. We have modified this word.

- 8: Please show the TIN mesh as well. Is it made of a single connected component?

(What if the 6 boreholes in the top left corner were further apart?)

Thank you for the suggestion. We have revised the graphic.

- 274: what is the "geological semantic information"? This is an important concept that needs to be properly defined.

"Geological semantic" is related to the relationships inherent in the borehole data. We have removed the ambiguous statement.

- 10: which cut lines?

We have modified the description in the updated paper: 10 shows the results of a straight cut along the S1 and S3 profile and a cut along the borehole."

- 290 "reasonable" is the real question: compared to what? Which features are you paying attention to? Can you get some quantitative measure? The comparison of figure 11 shows that the HRBF model is much more "reasonable" (except for the base)···

Thank you for your suggestion. We have detailed our criteria in the new Discussion section (4.3).

- 296: what is the "rationality"?

This was a lack of precision in our wording, and we have removed this vague expression.

- 303-310: the comparison needs to be more explicit about the better results from HRBF, instead of mentioning "certain differences"

Based on your suggestion, we have described the issue in detail in the new Discussion section (4.1).

- 326 "however" is the wrong logical link

Thank you for your suggestion. We have modified this section.

- 351, I don't understand what is the model "with close raster accuracy"

This was a lack of precision in our wording, and we have removed this vague expression.

- 13, colorbar legends of (b) and (c) are wrong

Thank you for your suggestion. We have modified this section.

- 346 the comparison is not all that clear from the figures... It could be true if focusing on the front face of the model, but not if looking at the top face. Compute the difference in the whole model and plot that difference to better illustrate. Does the same conclusion still hold?

Thank you for the suggestion. We have revised the graphic and have come to the same conclusion.

- 361: what do you mean by "artificial settings"?

This was a lack of precision in our wording, and we have removed this vague

expression.

- 370: which "predicted area"?

This was a lack of precision in our wording, and we have removed this vague expression.

- 374 "thicker boreholes"? or boreholes showing with thicker units?

This was a lack of precision in our wording, and we have removed this vague expression.

Thank you very much for your consideration.

Best regards!
Yours sincerely,
Jiateng Guo

---

## Author Response (AR2)

Dear Editor and reviewers,

we would like to thank you for your kind letter and for reviewers' constructive comments concerning our manuscript (**A Semisupervised Deep Learning Neural Network Using Pseudolabels for Three-Dimensional Shallow Strata Modelling and Uncertainty Analysis in Urban Areas from Borehole Data**). These comments are all valuable and helpful for improving our article. All the authors have seriously discussed about all these comments. According to the reviewers' comments, we have tried best to modify our manuscript to meet with the requirements of your journal. Some main modifications as follows:

(1) We have added to the reasons why we undertook this work and why we chose the ML approach in the new manuscript.

(2) We have added a table to the new manuscript that includes the network architecture and parameters to ensure reproducibility of our work.

(3) We have revised an ambiguous statement in line 71 of the introduction to the previous version of the manuscript

**Detailed modifications and responses are as follows:**

**Topic editor decision**

1) discuss the advantages and limitations of the proposed approach, to clarify explicitly the benefits of this ML approach compared to implicit methods. The purpose is not so much to provide an exhaustive comparison but to address the critical reason for this study in the first place, which is essential.

Thank you for your suggestion. We have included the following statement in lines 64-70 of the revised manuscript to explain why we launched this study and why the problem needs to be solved by the ML approach: "When using the implied surface method to construct a 3D geologic model…model by uncertainty analysis."

2) Ensure that all the details required are provided for reproducibility purposes (network architecture and parameters).

Thank you for your suggestion. We have added a detailed Table 2 to the revised manuscript on line 198 according to your and the other reviewers' suggestions.

**referee #1**

Line 17: What do you mean by "faces challenges related to uncertainty"? Implicit approaches are

more automated than explicit ones, so it should be easier to manage uncertainty. However, they struggle to capture complex structures (e.g., Collon et al., 2016, 10.1190/INT-2015-0178.1).

"Faces challenges related to uncertainty" means that constructing a 3D geological model by implicit approaches may lead to modeling that deviates from the real situation; we have modified this description in the Introduction section on lines 64-67.

Line 17: Kriging is one of the key approaches to implicit modeling, and kriging is the same as Gaussian processes in machine learning. So that opposition between explicit and implicit approaches on one side and machine learning approaches on the other is poorly substantiated. Machine learning is not a new tool for either explicit or implicit modeling. But using deep learning for explicit or implicit modeling is more recent, that would be true.

The main focus of this manuscript is to address the challenges of human labor consumption in explicit modeling and the inability to conduct uncertainty analysis in implicit modeling through machine learning methods. It is not intended to set machine learning modeling against explicit or implicit modeling methods.

Line 18-19: It's not so much the use of machine learning (see my comment above, Gaussian processes work well in data-poor settings), but the use of implicit modeling (see Collon et al., 2016, 10.1190/INT-2015-0178.1, although their case study has considerably less data than this one, so I'm not sure this study can be called "data-poor").

Indeed, this manuscript addresses different modeling data than the article you referenced. The focus of this manuscript is primarily on the challenges encountered in modeling borehole data.

Line 27: What does that mean "better [...] supports uncertainty analysis"?

We want to show that the SDLP algorithm is the best algorithm in the manuscript.

Line 29: This is not a case study with sparse borehole data. Having a borehole every 23 m or so covering most of a 305 by 264 m domain is a high data density for a subsurface project.

The manuscript presents the borehole data obtained from a real engineering project in Shenyang city. The primary objective of this project is to ensure the stability of the building.

Line 37: It is a bit weird to end a list of common data with "other types of data". That list is limited in subsurface projects, so better be exhaustive. Maybe analog data from outcrops could be worth adding.

Thank you for your suggestion. We have modified these problems according to your suggestion.

Line 54: What does "MLS" stand for?

Thank you for your suggestion. We have added this information to the lost section.

Line 70: I don't understand the first point.

We have modified this section.

Line 125: What if the TIN connect two intervals that shouldn't be connected? For instance two sandy intervals, but one is from a sedimentary channel, the other a crevasse splay? What would be the

Incorrectly connecting intervals in geological modeling significantly reduces model accuracy. The connectivity of geological layers is vital for ensuring accurate representations of actual geological conditions. Inaccurate connections can lead to models that fail to reflect the true geological characteristics. Moreover, such errors pose challenges in geological interpretation, especially in boundary areas, because they can affect the accurate understanding of lithology, structural features, and depositional environments. In the modeling process, the sedimentary sequence of the strata is used as the basis for connections between geological layers; therefore, it is believed that there are no issues associated with incorrect connections.

Figure 1: I'm still not sure I fully understand, so on the zoomed section, what are the points H1iPj mean? Are they just to illustrate the balance between different intervals? Or are they new data points created with the TIN? This remains quite confusing.

$H1iPj$ represents unequal-interval sampling performed on deterministic profiles, and its coordinates x, y, and z can be obtained using Formula (2).

Line 170: It would be nice to have a figure describing the network's architecture (this could be done by updating figure 2).

Thank you for your suggestion. We have modified Figure 2.

Line 170: How can a user choose the right number of hidden neurons? How many layers are used? 4 like in figure 2? Why that choice? Is it robust or does it impact predictions significantly?

The structure and hyperparameters of the neural network in this manuscript were primarily referenced from the article "www.zjujournals.com/eng/article/2021/1008-973X/202103021.shtml " and manually adjusted through experimentation. However, the selection of the neural network structure and determination of hyperparameters are not within the scope of this manuscript's research. The primary focus of this manuscript is to demonstrate the effectiveness of using pseudolabels in deep learning networks for constructing 3D geological models.

Line 195: Here a figure would help a lot.

Thank you for your suggestion. We will take your opinion into consideration in our next stage of research.

Line 199: High accuracy based on what criterion?

Thank you for your suggestion. We have added this explanation to the manuscript. A training accuracy of 90% is considered to indicate a high accuracy rate.

Line 200: Is that done on the prisms mentioned before? Actually I'm still confused.

"There might have been a misunderstanding; it is done on the unlabelled data. This statement means that pseudolabelled data are obtained by comparing the predicted results of the unlabelled data with the attributes of the prism where the coordinates of the unlabelled data reside."

Line 234: I'm still not sure what "unlabelled grids" mean. Are those related to the deterministic profiles?

"The unlabelled grids" mean that the grids except from borehole data and pseudolabel data.

We set the training set:validation set:test set to be 6:2:2, and we have added this detail to Table.2.

Table 2: What are SAM and DL? And where is HRBF?

"SAM" should be SVM. We apologize, but the HRBF algorithm does not fall under the category of machine learning methods. As a result, it does not provide outputs such as accuracy, precision, recall, or F1 score.

Figure 5 & 6: Is that only for SPDL? What about the other methods?

Because the manuscript focuses on the SDLP algorithm, we only show the results of the SDLP algorithm.

Yes, the grid size is the cell size.

Line 266: How was the K-fold set up? By removing individual labels or entire wells?

By removing entire wells.

Line 272: The procedure for the validation is not clear. You're only removing a few wells, so why not simply do a holdout validation (i.e., train only once to predict all the missing wells together directly)? And/or why not do a proper group K-fold cross-validation and predict all the wells?

In fact, the article uses K-fold cross-validation (with K set to 10) and provides the average results of cross-validation.

Line 277: But in practice you won't be able to make such choices. This adds a considerable bias to your validation.

The conditions under which this study was conducted were met when we already had some idea of the results. We will consider the research you mentioned regarding real-life situations as part of our future research plans.

Sections 4.1, 4.2, and 4.4: Those are not discussions, but new results that should be in section 3. The validation should be the same for all methods.

We have reorganized the sections of this manuscript in this way because we want to incrementally explain the effectiveness of our proposed SDLP algorithm.

Line 311: I see no real reason for that. The goal of this paper is to predict geological formations, and the HRBF method should be compared on the same ground.

This manuscript demonstrates the conformity between the drilling data and the established three-dimensional geological model through Fig. 11b-d.

Line 329: A visual comparison is not robust enough for this. You need to add a quantified validation, so include the HRBF method in section 3.

Thank you for your suggestion. Our current research is unable to address the issue you have raised. We will endeavor to conduct further investigations in our next phase of research.

 That's just untrue, erosive structures such as channels will lead to abrupt changes. You actually cannot say whether the HRBF method or SPDL perform better based on such weak comparison based on unsupported claims.

Thank you for bringing up this concern. We have carefully reviewed the geological reports of the area in question, and there is no evidence of an erosive structure, as you described.

 That claim only holds is the both algorithms were properly tuned, which is not mentioned in the paper.

The main focus of this manuscript is to highlight the effectiveness of the proposed SDLP algorithm. Throughout the main text, we have provided evidence to support the effectiveness of this algorithm.

 What about other indicators than accuracy? In imbalanced cases, accuracy is actually a poor indicator of performance since good accuracy can be achieved by predicting the major classes only.

We have considered your previous suggestion and have incorporated precision, recall, and F1 score as our evaluation metrics.

 However, other approaches to implicit modelling can capture uncertainty, how does you method compare to those?

Currently, our research has not included uncertainty comparisons with other implicit approaches. We will endeavor to supplement our study with this aspect in future work.

**referee #2**

• L.37 "and other type of data" not needed since listing "included" examples.

Thank you for your suggestion. We have removed "and other types of data".

• L.71 I don't understand "is much less than not revealed by borehole data", please rephrase

We have modified the fuzzy expression.

• L.74 "were mainly" -> do you mean "are usually"?

Thank you for your suggestion; we have modified the tense.

• L.75 it is the distribution of categories that's imbalanced, not their number

In the manuscript, "the imbalanced number of categories" is shown that is the difficult that these data are used for training data to training deep learning model.

• L.102, "maximum average thickness" -> per formation. The authors might omit that sentence altogether, as the average thickness does not seem to be used or referred to anywhere else.

Thank you for your suggestion; we have removed this term.

• L.124, please specify the (redundant but helpful) 2D horizontal nature of the TIN since talking

approximately 3D until then.

Thank you for your suggestion. We have added this explanation to the manuscript.

• L.128, please specify the measure and quantitative threshold used (skewness?) to remove "narrow triangles"

Thank you for your suggestion. We have added this explanation to the manuscript. The threshold for determining whether a triangle is an acute triangle based on the measurement of its smallest angle is set to 20 degrees.

• L.170: ReLU instead of RELU

Thank you for your suggestion. We have modified the spelling.

• L.171: what percentage is used for the dropout?

The dropout percentage is set to 0.1. We aimed to avoid excessive resetting of neurons, which could lead to incorrect predictions and erroneous pseudolabelled data.

• Fig.2 "pridiction" typo

Thank you for your suggestion. We have modified the spelling.

Thank you very much for your consideration.

Best regards!
Yours sincerely,
Jiateng Guo